# Functional Regularization for Representation Learning: A Unified Theoretical Perspective

**Siddhant Garg**[*]
Amazon Alexa AI Search
Manhattan Beach, CA, USA
sidgarg@amazon.com

**Yingyu Liang**
Department of Computer Sciences
University of Wisconsin-Madison
yliang@cs.wisc.edu

## Abstract

Unsupervised and self-supervised learning approaches have become a crucial tool to learn representations for downstream prediction tasks. While these approaches are widely used in practice and achieve impressive empirical gains, their theoretical understanding largely lags behind. Towards bridging this gap, we present a unifying perspective where several such approaches can be viewed as imposing a regularization on the representation via a *learnable* function using unlabeled data. We propose a discriminative theoretical framework for analyzing the sample complexity of these approaches, which generalizes the framework of [3] to allow learnable regularization functions. Our sample complexity bounds show that, with carefully chosen hypothesis classes to exploit the structure in the data, these learnable regularization functions can prune the hypothesis space, and help reduce the amount of labeled data needed. We then provide two concrete examples of functional regularization, one using auto-encoders and the other using masked self-supervision, and apply our framework to quantify the reduction in the sample complexity bound of labeled data. We also provide complementary empirical results to support our analysis.

## 1 Introduction

Advancements in machine learning have resulted in large prediction models, which need large amounts of labeled data for effective learning. Expensive label annotation costs have increased the popularity of unsupervised (or self-supervised) representation learning techniques using additional unlabeled data. These techniques learn a representation function on the input, and a prediction function over the representation for the target prediction task. Unlabeled data is utilised by posing an auxiliary unsupervised learning task on the representation, e.g., using the representation to reconstruct the input. Some popular examples of the auxiliary task are auto-encoders [45, 4], sparse dictionaries [46], masked self-supervision [13], manifold learning [11], among others [9]. These approaches have been extensively used in applications in domains such as computer vision (e.g., [56, 61, 16]) and natural language processing (e.g., [58, 13, 34]), and have achieved impressive empirical performance.

An important takeaway from these empirical studies is that learning representations using unlabeled data can drastically reduce the size of labeled data needed for the prediction task. In contrast to the popularity and impressive practical gains of these representation learning approaches, there have been far fewer theoretical studies focused at understanding them, most of which have been specific to individual approaches. While intuition dictates that the unlabeled and labeled data distributions along with the choice of models are crucial factors which govern the empirical gains, theoretically there is still ambiguity over questions like *"When can the auxiliary task over the unlabeled data help the target prediction task? How much can it reduce the sample size of the labeled data by?"*

---

[*] Work completed at the University of Wisconsin-Madison

In this paper, we take a step towards improving the theoretical understanding of the benefits of learning representations for the target prediction task via an auxiliary task. We focus on analyzing the sample complexity of labeled and unlabeled data for this learning paradigm. Such an analysis can help to identify conditions when a significant reduction in sample complexity of the labeled data can be achieved. Arguably, this is one of the most fundamental questions for this learning paradigm, and existing literature on this has been limited and scattered, specific to individual approaches.

Our contribution is to propose a unified perspective where several representation learning approaches can be viewed as if they impose a regularization on the representation via a learnable regularization function. Under this paradigm, representations are learned jointly on unlabeled and labeled data. The former is used in the auxiliary task to learn the representation and the regularization function. The latter is used in the target prediction task to learn the representation and the prediction function. Henceforth, we refer to this paradigm as *representation learning via functional regularization*.

In particular, we present a PAC-style discriminative framework [53] to bound the sample complexities of labeled and unlabeled data under different assumptions on the models and data distributions. This is inspired from the work of [3] which bounds the sample complexities of labeled and unlabeled data for semi-supervised learning. Our generalized framework allows *learnable* regularization functions and thus unifies multiple unsupervised (or self-supervised) representation learning approaches. Our analysis shows that functional regularization with unlabeled data can prune the model hypothesis class for learning representations, reducing the labeled data required for the prediction task.

To demonstrate the application of our framework, we construct two concrete examples of functional regularization, one using auto-encoder and the other using masked self-supervision. These specific functional regularization settings allow us to quantify the reduction in the sample bounds of labeled data more explicitly. While our main focus is the theoretical framework, we also provide complementary empirical support through experiments on synthetic and real data. Now we first discuss related work followed by a formal problem description. Then we present our theoretical framework involving sample complexity bounds followed by the concrete examples with empirical support.

## 2   Related Work

Self-supervised learning approaches for images have been extensively used in computer vision through auxiliary tasks such as masked image patch prediction [16], image rotations [20], pixel colorization [60], context prediction of image patches [14, 41, 39, 26], etc. Additionally, variants of these approaches find practical use in the field of robotics [49, 42, 1, 18, 27]. Masked self-supervision (a type of denoising auto-encoder), where representations are learnt by hiding a portion of the input and then reconstructing it, has lead to powerful language models like BERT [13] and RoBERTa [34] in natural language processing. There have also been numerous studies on other representation learning approaches such as RBMs [29, 30], dictionary learning [44, 35] and manifold learning [43]; [9] presents an extensive review of multiple representation learning approaches.

On the theoretical front, [3] presents a discriminative framework for analyzing semi-supervised learning showing that unlabeled data can reduce the labeled sample complexity. Our framework in this paper is inspired from [3], and generalizes their analysis for utilizing unlabeled data through a *learnable* regularization function. This allows a unified theoretical framework to study multiple representation learning approaches. In addition to [3], [12] also studies the benefits of using unlabeled data, but by restricting that the unlabeled data be utilized through a fixed function. Some other works [7, 8] have explored the benefits of unlabeled data for domain adaptation. Our setting differs from this since our goal is to learn a prediction function on the labeled data, rather than for a change in the domain of labeled data from source to target. Another line of related work considers multi-task learning, such as [6, 33]. These works show that multiple supervised learning tasks on different, but related, data distributions can help generalization. Our work differs from these since we focus on learning a supervised task using auxiliary unsupervised tasks on unlabeled data.

[19] presents a comprehensive empirical study on the benefits of unsupervised pre-training for image-classification tasks in computer vision. Our analysis in this paper is motivated by their empirical results showing that pre-training shrinks the hypothesis space searched during learning. There have also been theoretical studies on several representation approaches individually, without providing a holistic perspective. [47] presents a theoretical framework to analyse unsupervised representation learning techniques that can be posed as a contrastive learning problem, with their

results later improved by [40]. [23] provide a theoretical analysis of unsupervised learning from an optimization viewpoint, with applications to dictionary learning and spectral auto-encoders. [31] prove uniform stability generalization bounds for linear auto-encoders and empirically demonstrate the benefits of using supervised auto-encoders. Additionally, there are some studies on learning transferable representations using multiple tasks [17, 51, 22]. Another line of related work includes approaches [2, 52] that analyze representation learning from the perspective of maximizing the mutual information between the data and the representation. Connecting these mutual information approaches with our framework is left as future work.

## 3  Problem Formulation

Consider labeled data $S = \{(x_i, y_i)\}_{i=1}^{m_\ell}$ from a distribution $\mathcal{D}$ over the domains $\mathcal{X} \times \mathcal{Y}$, where $\mathcal{X} \subseteq \mathbb{R}^d$ is the input feature space and $\mathcal{Y}$ is the label space. The goal is to learn a predictor $p : \mathcal{X} \to \mathcal{Y}$ that fits $\mathcal{D}$. This can be achieved by first learning a representation function $\phi = h(x) \in \mathbb{R}^r$ over the input, and then learning a predictor $y=f(\phi)\in\mathcal{Y}$ on the representation. Denote the hypothesis classes for $h$ and $f$ by $\mathcal{H}$ and $\mathcal{F}$ respectively, and the loss function by $\ell_c(f(h(x)), y)$. Without loss of generality, we assume $\ell_c \in [0, 1]$. We are interested in representation learning approaches where $h(x)$ is learned with the help of an auxiliary task on unlabeled data $U = \{\tilde{x}_i\}_{i=1}^{m_u}$ from a distribution $\mathcal{U}_X$ (same or different from the marginal distribution $\mathcal{D}_X$ of $\mathcal{D}$). Such an auxiliary task is posed as an unsupervised (or rather a self-supervised) learning task depending only on the input feature $x$.

Representation learning using auto-encoders is an example that fits this consideration, where given input $x$, the goal is to learn $\phi = h(x)$ s.t. $x$ can be decoded back from $h(x)$. More precisely, the decoder $d$ takes the representation $\phi$ and decodes it to $\hat{x}=g(\phi) \in \mathbb{R}^d$. $h$ and $g$ are learnt by minimizing the reconstruction error between $\hat{x}$ and $x$ (e.g., $\|x - \hat{x}\|_2=\|x-g(h(x))\|_2$). Representation learning via masked self-supervision is another example of our problem setting, where the goal is to learn $\phi = h(x)$ such that a part of the input (e.g., $x_1$) can be predicted from the representation of the remaining part (e.g., $h(x')$ on input $x' = [0, x_2, \ldots, x_d]$ with $x_1$ masked). This approach uses a function $g$ over the representation to predict the masked part of the input. $h$ and $g$ are optimized by minimizing the error between the true and the predicted values (e.g., $(x_1-g(h(x')))^2$).

Now consider a simple example of a regression problem where $y=\sum_{i=1}^d x_i$ and we use masked self-supervision to learn $x_1$ from $x'$. If each $x_i\sim\mathcal{N}(0,1)$ $i.i.d.$, then $h$ will not be able to learn a meaningful representation of $x$ for predicting $y$, since $x_1$ is independent of all other coordinates of $x$. On the other extreme, if all $x_i$'s are equal, $h$ can learn the perfect underlying representation for predicting $y$, which corresponds to a single coordinate of $x$. This shows two contrasting abstractions of the inherent structure in the data and how the benefits of using a specific auxiliary task may vary. Our work aims at providing a framework for analyzing sample complexity and clarifying these subtleties on the benefits of the auxiliary task depending on the data distribution.

## 4  Functional Regularization: A Unified Perspective

We make a key observation that the auxiliary task in several representation learning approaches provides a regularization on the representation function via a learnable function. To better illustrate this viewpoint, consider the auxiliary task of an auto-encoder, where the decoder $g(\phi)$ can be viewed as such a learnable function, and the reconstruction error $\|x - g(h(x))\|_2$ can be viewed as a regularization penalty imposed on $h$ through the decoder $g$ for the data point $x$.

To formalize this notion, we consider learning representations via an auxiliary task which involves: a learnable function $g$, and a loss of the form $L_r(h, g; x)$ on the representation $h$ via $g$ for an input $x$. We refer to $g$ as the regularization function and $L_r$ as the regularization loss. Let $\mathcal{G}$ denote the hypothesis class for $g$. Without loss of generality we assume that $L_r \in [0, 1]$.

**Definition 1.** Given a loss function $L_r(h, g; x)$ for an input $x$ involving a representation $h$ and a regularization function $g$, the regularization loss of $h$ and $g$ on a distribution $\mathcal{U}_X$ over $\mathcal{X}$ is defined as

$$L_r(h, g \,; \mathcal{U}_X) := \mathbb{E}_{x\sim\mathcal{U}_X} \left[ L_r(h, g; x) \right]. \tag{1}$$

The regularization loss of a representation function $h$ on $\mathcal{U}_X$ is defined as

$$L_r(h \,; \mathcal{U}_X) := \min_{g\in\mathcal{G}} L_r(h, g \,; \mathcal{U}_X). \tag{2}$$

We can similarly define $L_r(h, g\,;U)$ and $L_r(h\,;U)$ to denote the loss over a fixed set $U$ of unlabeled data points, i.e., $L_r(h, g\,;U) := \frac{1}{|U|} \sum_{x \in U} L_r(h, g\,;x)$ and $L_r(h\,;U) := \min_{g \in \mathcal{G}} L_r(h, g\,;U)$.

Here, $L_r(h\,;\mathcal{U}_X)$ can be viewed as a notion of incompatibility of a representation function $h$ on the data distribution $\mathcal{U}$. This formalizes the prior knowledge about the representation function and the data. For example, in auto-encoders $L_r(h\,;\mathcal{U}_X)$ measures how well the representation function $h$ complies with the prior knowledge of the input being reconstructible from the representation.

We now introduce a notion for the subset of representation functions having a bounded regularization loss, which is crucial for our sample complexity analysis.

**Definition 2.** Given $\tau \in [0, 1]$, the $\tau$-regularization-loss subset of representation hypotheses $\mathcal{H}$ is:

$$\mathcal{H}_{\mathcal{D}_X, L_r}(\tau) := \{h \in \mathcal{H} : L_r(h\,;\mathcal{D}_X) \leq \tau\}. \tag{3}$$

We also define the prediction loss over the data distribution $\mathcal{D}$ for a prediction function $f$ on top of $h$: $L_c(f, h\,;\mathcal{D}) := \mathbb{E}_{(x,y) \sim \mathcal{D}} [\ell_c(f(h(x)), y)]$, where $\ell_c$ is the loss function for prediction defined in Section 3. Similarly, the empirical loss on the labeled data set $S$ is $L_c(f, h\,;S) := \frac{1}{|S|} \sum_{(x,y) \in S} \ell_c(f(h(x)), y)$. In summary, given hypothesis classes $\mathcal{H}, \mathcal{F}$, and $\mathcal{G}$, a labeled dataset $S$, an unlabeled dataset $U$, and a threshold $\tau > 0$ on the regularization loss, we consider the following learning problem:

$$\min_{f \in \mathcal{F}, h \in \mathcal{H}} L_c(f, h\,;S), \ \text{ s.t. } L_r(h\,;U) \leq \tau. \tag{4}$$

### 4.1 Instantiations of Functional Regularization

Here we show some popular representation learning approaches as instantiations of our framework by specifying the analogous regularization functions $\mathcal{G}$ and regularization losses $L_r$. We mention several other instantiations of our framework like manifold learning, dictionary learning, etc in Appendix A.

**Auto-encoder.** Recall from Section 3, there is a decoder function $g$ that takes the representation $h(x)$ and decodes it to $\hat{x} = g(h(x))$, where $h$ and $g$ are learnt by minimizing the error $\|x - \hat{x}\|_2$. The reconstruction error corresponds to the regularization loss $L_r(h, g\,;x)$. $\mathcal{H}_{\mathcal{D}_X, L_r}(\tau)$ is the subset of representation functions with at most $\tau$ reconstruction error using the best decoder in $\mathcal{G}$.

**Masked Self-supervised Learning.** Recall the simple example from Section 3 where $x_1$, the first dimension of input $x$, is predicted from the representation $h(x')$ of the remaining part (i.e., $x' = [0, x_2, \ldots, x_d]$ with $x_1$ masked). The prediction function for $x_1$ using $h(x')$ corresponds to the regularization function $g$, and $\|x_1 - g(h(x'))\|_2^2$ corresponds to the regularization loss $L_r(h, g\,;x)$. $\mathcal{H}_{\mathcal{D}_X, L_r}(\tau)$ is the subset of $\mathcal{H}$ which have at most $\tau$ MSE on predicting $x_1$ using the best function $g$. More general denoising auto-encoder methods can be similarly mapped in our framework.

**Variational Auto-encoder.** VAEs encode the input $x$ as a distribution $q_\phi(z|x)$ over a parametric latent space $z$ instead of a single point, and sample from it to reconstruct $x$ using a decoder $p_\theta(x|z)$. The encoder $q_\phi(z|x)$ is used to model the underlying mean $\mu_z$ and co-variance matrix $\sigma_z$ of the distribution over $z$. VAEs are trained by minimising a loss over parameters $\theta$ and $\phi$

$$\mathcal{L}_x(\theta, \phi) = -\mathbb{E}_{z \sim q_\phi(z|x)}[\log\ p_\theta(x|z)] + \mathbb{KL}(q_\phi(z|x)||p(z))$$

where $p(z)$ is specified as the prior distribution over $z$ (e.g., $\mathcal{N}(0, 1)$). The encoder $q_\phi(z|x)$ can be viewed as the representation function $h$, the decoder $p_\theta(x|z)$ as the learnable regularization function $g$, and the loss $\mathcal{L}_x(\theta, \phi)$ as the regularization loss $L_r(h, g;x)$ in our framework. Then $\mathcal{H}_{\mathcal{D}_X, L_r}(\tau)$ is the subset of encoders $q_\phi$ which have at most $\tau$ VAE loss when using the best decoder $p_\theta$ for it.

### 4.2 Sample Complexity Analysis

To analyze the sample complexity of representation learning via functional regularization, we generalize the analysis from [3] by extending it from semi-supervised learning with unlabeled data using a fixed regularization function to the general setting of using *learnable* regularization functions with unlabeled data. We first enumerate the different considerations on the data distributions and the hypothesis classes: 1) the labeled and unlabeled data can either be from the same or different distributions (i.e., same domain or different domains); 2) the hypothesis classes can contain zero

error hypothesis or not (i.e., being realizable or unrealizable); 3) the hypothesis classes can be finite or infinite in size. We perform sample complexity analysis for different combinations of these assumptions. While the bounds across different settings have some differences, the proofs share a common high-level intuition. We now present sample complexity bounds for three interesting, characteristic settings. We present bounds for several other settings along with proofs of all the theorems in Appendix B.

**Same Domain, Realizable, Finite Hypothesis Classes.** We start with the simplest setting, where the unlabeled dataset $U$ and the labeled dataset $S$ are from the same distribution $\mathcal{D}_X$, and the hypothesis classes $\mathcal{F}, \mathcal{G}, \mathcal{H}$ contain functions $f^*, g^*, h^*$ with a zero prediction and regularization loss. We further assume that the hypothesis classes are finite in size. We can prove the following result:

**Theorem 1.** *Suppose there exist $h^* \in \mathcal{H}, f^* \in \mathcal{F}, g^* \in \mathcal{G}$ such that $L_c(f^*, h^*; \mathcal{D}) = 0$ and $L_r(h^*, g^*; \mathcal{D}_X) = 0$. For any $\epsilon_0, \epsilon_1 \in (0, 1/2)$, a set $U$ of $m_u$ unlabeled examples and a set $S$ of $m_l$ labeled examples are sufficient to learn to an error $\epsilon_1$ with probability $1 - \delta$, where*

$$m_u \geq \frac{1}{\epsilon_0} \left[ \ln |\mathcal{G}| + \ln |\mathcal{H}| + \ln \frac{2}{\delta} \right], \qquad m_l \geq \frac{1}{\epsilon_1} \left[ \ln |\mathcal{F}| + \ln |\mathcal{H}_{\mathcal{D}_X, L_r}(\epsilon_0)| + \ln \frac{2}{\delta} \right]. \quad (5)$$

*In particular, with probability at least $1 - \delta$, all hypotheses $h \in \mathcal{H}, f \in \mathcal{F}$ with $L_c(f, h; S) = 0$ and $L_r(h; U) = 0$ will have $L_c(f, h; \mathcal{D}) \leq \epsilon_1$.*

Theorem 1 shows that, if the target function $f^* \circ h^*$ is indeed perfectly correct for the target prediction task, and $h^* \circ g^*$ has a zero regularization loss, then optimizing the prediction and regularization loss to 0 over the above mentioned number of data points (Equation 5) is sufficient to learn an accurate predictor in the PAC learning sense.

Recall that PAC analysis for the standard setting of the prediction task only using labeled data (and no unlabeled data) shows that $\frac{1}{\epsilon_1} \left[ \ln |\mathcal{F}| + \ln |\mathcal{H}| + \ln \frac{2}{\delta} \right]$ labeled points are needed to get the same error guarantee. On comparing the bounds, Theorem 1 shows that the functional regularization can prune away some hypotheses in $\mathcal{H}$; thereby replacing the factor $\mathcal{H}$ with its subset $\mathcal{H}_{\mathcal{D}_X, L_r}(\epsilon_0)$ in the bound. Thus, the sample complexity bound is reduced by $\frac{1}{\epsilon_1} [\ln |\mathcal{H}| - \ln |\mathcal{H}_{\mathcal{D}_X, L_r}(\epsilon_0)|]$. Equivalently, the error is reduced by $\frac{1}{m_\ell} [\ln |\mathcal{H}| - \ln |\mathcal{H}_{\mathcal{D}_X, L_r}(\epsilon_0)|]$ when using $m_\ell$ labeled data. So the auxiliary task is helpful for learning the predictor when $\mathcal{H}_{\mathcal{D}_X, L_r}(\epsilon_0)$ is significantly smaller than $\mathcal{H}$, avoiding the requirement of a large number of labeled points to find a good representation function among them.

**Same Domain, Unrealizable, Infinite Hypothesis Classes.** We now present the result for a more elaborate setting, where both the prediction and regularization losses are non-zero. We also relax the assumptions on the hypothesis classes being finite. We use metric entropy to measure the capacity of the hypothesis classes for demonstration here. Alternative capacity measures like VC-dimension or Rademacher complexity can also be used with essentially no change to the analysis. Assume that the parameter space of $\mathcal{H}$ is equipped with a norm and let $\mathcal{N}_{\mathcal{H}}(\epsilon)$ denote the $\epsilon$-covering number of $\mathcal{H}$; similarly for $\mathcal{F}$ and $\mathcal{G}$. Let the Lipschitz constant of the losses w.r.t. these norms be bounded by $L$.

**Theorem 2.** *Suppose there exist $h^* \in \mathcal{H}, f^* \in \mathcal{F}, g^* \in \mathcal{G}$ such that $L_c(f^*, h^*; \mathcal{D}) \leq \epsilon_c$ and $L_r(h^*, g^*; \mathcal{D}_X) \leq \epsilon_r$. For any $\epsilon_0, \epsilon_1 \in (0, 1/2)$, a set $U$ of $m_u$ unlabeled examples and a set $S$ of $m_l$ labeled examples are sufficient to learn to an error $\epsilon_c + \epsilon_1$ with probability $1 - \delta$, where*

$$m_u \geq \frac{C}{\epsilon_0^2} \ln \frac{1}{\delta} \left[ \ln \mathcal{N}_{\mathcal{G}} \left( \frac{\epsilon_0}{4L} \right) + \ln \mathcal{N}_{\mathcal{H}} \left( \frac{\epsilon_0}{4L} \right) \right], \qquad (6)$$

$$m_l \geq \frac{C}{\epsilon_1^2} \ln \frac{1}{\delta} \left[ \ln \mathcal{N}_{\mathcal{F}} \left( \frac{\epsilon_1}{4L} \right) + \ln \mathcal{N}_{\mathcal{H}_{\mathcal{D}_X, L_r}(\epsilon_r + 2\epsilon_0)} \left( \frac{\epsilon_1}{4L} \right) \right] \qquad (7)$$

*for some absolute constant $C$. In particular, with probability at least $1 - \delta$, the $h \in \mathcal{H}, f \in \mathcal{F}$ that optimize $L_c(f, h; S)$ subject to $L_r(h; U) \leq \epsilon_r + \epsilon_0$ have $L_c(f, h; \mathcal{D}) \leq L_c(f^*, h^*; \mathcal{D}) + \epsilon_1$.*

Theorem 2 shows that optimizing the prediction loss subject to the regularization loss bounded by $(\epsilon_r + \epsilon_0)$ can give a solution $f \circ h$ with prediction loss close to the optimal. The sample complexity bounds are broadly similar to those in the simpler realizable and finite hypothesis class setting, but with $\frac{1}{\epsilon_0}$ and $\frac{1}{\epsilon_1}$ replaced by $\frac{1}{\epsilon_0^2}$ and $\frac{1}{\epsilon_1^2}$ due to the unrealizability, and logarithms of the hypothesis class sizes replaced by their metric entropy due to the classes being infinite. We show a reduction of $\frac{C}{\epsilon_1^2} \left[ \ln \mathcal{N}_{\mathcal{H}} \left( \frac{\epsilon_1}{4L} \right) - \ln \mathcal{N}_{\mathcal{H}_{\mathcal{D}_X, L_r}(\epsilon_r + 2\epsilon_0)} \left( \frac{\epsilon_1}{4L} \right) \right]$ with the standard bound on $m_l$ without unlabeled data. Equivalently, the error bound is reduced by $\frac{C}{\sqrt{m_\ell}} \left[ \ln \mathcal{N}_{\mathcal{H}} \left( \frac{\epsilon_1}{4L} \right) - \ln \mathcal{N}_{\mathcal{H}_{\mathcal{D}_X, L_r}(\epsilon_r + 2\epsilon_0)} \left( \frac{\epsilon_1}{4L} \right) \right]$.

We bring attention to some subtleties which are worth noting. Firstly, the regularization loss $\epsilon_r$ of $g^*, h^*$ need not be optimal; there may be other $g, h$ which get a smaller $L_r(h, g; \mathcal{D}_X)$ (even $\ll \epsilon_r$). Secondly, the prediction loss is bounded by $L_c(f^*, h^*; \mathcal{D}) + \epsilon_1$, which is independent of $\epsilon_r$. Similarly, the bounds on $m_u$ and $m_\ell$ mainly depend on $\epsilon_0$ and $\epsilon_1$ respectively, while only $m_\ell$ depends on $\epsilon_r$ through the $\mathcal{H}_{\mathcal{D}_X, L_r}(\epsilon_r + 2\epsilon_0)$ term. Thus, even when the regularization loss is large (e.g., the reconstruction of an auto-encoder is far from accurate), it is still possible to learn an accurate predictor with a significantly reduced labeled data size using the unlabeled data. This suggests that when designing an auxiliary task ($\mathcal{G}$ and $L_r$), it is *not* necessary to ensure that the "ground-truth" $h^*$ has a small regularization loss. Rather, one should ensure that only a small fraction of $h \in \mathcal{H}$ have a smaller (or similar) regularization loss than $h^*$ so as to reduce the label sample complexity.

This bound also shows that $\tau$ should be carefully chosen for the constraint $L_r(h; U) \le \tau$. With a very small $\tau$, the ground-truth $h^*$ (or hypotheses of similar quality) may not satisfy the constraint and become infeasible for learning. With a very large $\tau$, the auxiliary task may not reduce the labeled sample complexity. Practical learning algorithms typically turn this constrain into a regularization like term, i.e., by optimizing $L_c(f, h; S) + \lambda L_r(h; U)$. For such objectives, the requirement on $\tau$ translates to carefully choosing $\lambda$. When $\lambda$ is very large, this leads to a small $L_r(h; U)$ but a large $L_c(f, h; S)$, while when $\lambda$ is very small, this may not reduce the labeled sample complexity.

**Different Domain, Unrealizable, Infinite Hypothesis Classes.** In practice, the unlabeled data is often from a different domain than the labeled data. For example, state-of-the-art NLP systems are often pre-trained on a large general-purpose unlabeled corpus (e.g., the entire Wikipedia) while the target task has a small specific labeled corpus (e.g., a set of medical records). That is, the unlabeled data $U$ is from a distribution $\mathcal{U}_X$ different from $\mathcal{D}_X$. For this setting, we have the following bound:

**Theorem 3.** *Suppose the unlabeled data $U$ is from a distribution $\mathcal{U}_X$ different from $\mathcal{D}_X$. Suppose there exist $h^* \in \mathcal{H}, f^* \in \mathcal{F}, g^* \in \mathcal{G}$ such that $L_c(f^*, h^*; \mathcal{D}) \le \epsilon_c$ and $L_r(h^*, g^*; \mathcal{U}_X) \le \epsilon_r$. Then the same sample complexity bounds as in Theorem 2 hold (replacing $\mathcal{D}_X$ with $\mathcal{U}_X$ in Equation 7).*

The bound is similar to that in the setting of the domain distributions being same. It implies that unlabeled data from a domain different than the labeled data, can help in learning the target task, as long as there exists a "ground-truth" representation function $h^*$, which is shared across the two domains, having a small prediction loss on the labeled data and a suitable regularization loss on the unlabeled data. The former (small prediction loss) is typically assumed according to domain knowledge, e.g., for image data, common visual perception features are believed to be shared across different types of images. The latter (suitable regularization loss) means only a small fraction of $h \in \mathcal{H}$ have a smaller (or similar) regularization loss than $h^*$, which requires a careful design of $\mathcal{G}$ and $L_r$.

### 4.3 Discussions

**When is functional regularization not helpful?** In addition to demonstrating the benefits of unlabeled data for the target prediction task, our theorems and analysis also provide implications for cases when the auxiliary self-supervised task may *not* help the target prediction task. Firstly, the regularization may not lead to a significant reduction in the size of the hypothesis class. For example, consider Theorem 1, if $\mathcal{H}_{\mathcal{D}_X, L_r}(\epsilon_0)$ is not significantly smaller than $\mathcal{H}$, then using unlabeled data will not reduce the sample size of the labeled data by much when compared to the case of only using labeled data for prediction. In fact, to get significant gain in sample complexity reduction, the size of the regularized hypothesis class $\mathcal{H}_{\mathcal{D}_X, L_r}(\epsilon_0)$ needs to be exponentially smaller than entire class $\mathcal{H}$. A polynomially smaller $\mathcal{H}_{\mathcal{D}_X, L_r}(\epsilon_0)$ only leads to minor logarithmic reduction in the sample complexity. Section 5 presents two concrete examples where the regularized hypothesis class is exponentially smaller than $\mathcal{H}$ thereby showing benefits of using functional regularization, but this can also help to identify examples where the auxiliary task does not aid learning. Secondly, the auxiliary task can fail if the regularization loss threshold ($\tau$ in Equation (4)) is not properly set. For example, if $\tau$ is set too small, then the feasible set ($\mathcal{H}_{\mathcal{D}_X, L_r}(\tau)$) may contain no hypotheses with a small prediction loss. Lastly, another possible reason that these representation learning approaches may fail is the inability of the optimization to lead to a good solution. Analyzing the effects of optimization for function regularization is an interesting future direction.

**Is uniform convergence suitable for our analysis?** Our sample complexity analysis is based on uniform convergence bounds. A careful reader may question whether uniform convergence is suitable for analyzing the generalization in the first place, since there is evidence [59, 37] that naïvely applying

uniform convergence bounds may not result in good generalization/sample bounds for deep learning. However, these existing studies [59, 37] cannot be directly applied to our problem setting. To the best of our knowledge, they apply for supervised learning tasks without any auxiliary representation learning on unlabeled data, which differs from our setting of using auxiliary tasks on unlabeled data. This difference in problem settings is the key in making uniform convergence bounds meaningful. More precisely, in supervised deep learning without auxiliary tasks, it is generally believed that the hypothesis class is larger than statistically necessary, and the optimization has an implicit regularization while training, and hence uniform convergence fails to explain the generalization (e.g., [38, 32]). However, in our setting with the auxiliary tasks, functional regularization has a regularization effect of restricting the learning to a smaller subset of the hypothesis space, as shown by our analysis and supported by empirical evidence in existing works (e.g., [19]) and our experiments in Section 6 and Appendix E. Once regularized to a smaller subset of hypotheses, the implicit regularization of the optimization is no longer significant, and thus the generalization can be explained by uniform convergence. A more thorough investigation is left as future work.

## 5   Applying the Theoretical Framework to Concrete Examples

The analysis in Section 4 shows that the sample complexity bound reduction depends on the notion of the pruned subset $\mathcal{H}_{\mathcal{D}_X, L_r}$, which captures the effect of the regularization function and the property of the unlabeled data distribution. Our generic framework can be applied to various concrete configurations of hypothesis classes and data distributions. This way we can quantify the reduction more explicitly by investigating $\mathcal{H}_{\mathcal{D}_X, L_r}$. We provide two such examples: one using an auto-encoder regularization and the other using a masked self-supervision regularization. We outline how to bound the sample complexity for these examples, and present the complete details and proofs in Appendix C.

### 5.1   An Example of Functional Regularization via Auto-encoder

**Learning Without Functional Regularization.**  Consider $\mathcal{H}$ to be the class of linear functions from $\mathbb{R}^d$ to $\mathbb{R}^r$ where $r < d/2$, and $\mathcal{F}$ to be the class of linear functions over some activations. That is,

$$\phi = h_W(x) = Wx, \ \ y = f_a(\phi) = \sum_{i=1}^r a_i \sigma(\phi_i) \, , \ \text{where } W \in \mathbb{R}^{r \times d}, \ \ a \in \mathbb{R}^r \qquad (8)$$

Here $\sigma(t)$ is an activation function (e.g., $\sigma(t){=}t^2$), the rows of $W$ and $a$ have $\ell_2$ norms bounded by 1. We consider the Mean Square Error prediction loss, i.e., $L_c(f, h; x){=}\|y - f(h(x))\|_2^2$. Without prior knowledge on data, no functional regularization corresponds to end-to-end training on $\mathcal{F} \circ \mathcal{H}$.

**Data Property.**    We consider a setting where the data has properties which allows functional regularization. We assume that the data consists of a signal and noise, where the signal lies in a certain $r$-dimensional subspace. Formally, let columns of $B \in \mathbb{R}^{d \times d}$ be eigenvectors of $\Sigma{:=}\mathbb{E}[xx^\top]$, then the prediction labels are largely determined by the signal in the first $r$ directions: $y{=}\sum_{i=1}^r a_i^* \sigma(\phi_i^*){+}\nu$ and $\phi^*{=}B_{1:r}^\top x$, where $a^* \in \mathbb{R}^r$ is a ground-truth parameter with $\|a^*\|_2{\leq}1$, $B_{1:r}$ is the set of first $r$ eigenvectors of $\Sigma$, and $\nu$ is a small Gaussian noise. We assume a difference in the $r^{\text{th}}$ and $r{+}1^{\text{th}}$ eigenvalues of $\Sigma$ to distinguish the corresponding eigenvectors. Let $\epsilon_r$ denote $\mathbb{E}\|x - B_{1:r} B_{1:r}^\top x\|_2^2$.

**Learning With Functional Regularization.**    Knowing that the signal lies in an $r$-dimensional subspace, we can perform auto-encoder functional regularization. Let $\mathcal{G}$ be a class of linear functions from $\mathbb{R}^r$ to $\mathbb{R}^d$, i.e., $\hat{x}{=}g_V(\phi){=}V\phi$ where $V \in \mathcal{R}^{d \times r}$ has orthonormal columns. The regularization loss $L_r(h, g; x){=}\|x - g(h(x))\|_2^2$. For simplicity, we assume access to infinite unlabeled data.

Without regularization, the standard $\epsilon$-covering argument shows that the labeled sample complexity, for an error $\epsilon$ close to the optimal, is $\frac{C}{\epsilon^2}\left[\ln \mathcal{N}_{\mathcal{F}}\left(\frac{\epsilon}{4L}\right) + \ln \mathcal{N}_{\mathcal{H}}\left(\frac{\epsilon}{4L}\right)\right]$ for some absolute constant $C$. Applying our framework when using regularization with $\tau = \epsilon_r$, the sample complexity is bounded by $\frac{C}{\epsilon^2}\left[\ln \mathcal{N}_{\mathcal{F}}\left(\frac{\epsilon}{4L}\right) + \ln \mathcal{N}_{\mathcal{H}_{\mathcal{D}_X, L_r}(\epsilon_r)}\left(\frac{\epsilon}{4L}\right)\right]$. Then we show that $\mathcal{N}_{\mathcal{H}}\left(\frac{\epsilon}{4L}\right) {\geq} \binom{d-r}{r} \mathcal{N}_{\mathcal{H}_{\mathcal{D}_X, L_r}(\epsilon_r)}\left(\frac{\epsilon}{4L}\right)$ (Proof in Lemma 6 of Appendix C.1) since

$$\mathcal{H}_{\mathcal{D}_X, L_r}(\epsilon_r) = \{h_W(x) : W = OB_{1:r}^\top, O \in \mathbb{R}^{r \times r}, O \text{ is orthonormal}\},$$

$$\mathcal{H} \supseteq \{h_W(x) : W = OB_S^\top, O \in \mathbb{R}^{r \times r}, O \text{ is orthonormal}, S \subseteq \{r+1, \ldots, d\}, |S|{=}r\},$$

where $B_S$ refers to the sub-matrix of columns in $B$ having indices in $S$. Therefore, the label sample complexity bound is reduced by $\frac{C}{\epsilon^2}\ln\binom{d-r}{r}$, i.e., the error bound is reduced by $\frac{C}{\sqrt{m_\ell}}\ln\binom{d-r}{r}$ when using $m_\ell$ labeled points. Note that $\ln\binom{d-r}{r}=\Theta(r\ln(d/k))$ when $r$ is small, and thus the reduction is roughly linear initially and then grows slower with $r$. Interestingly, the reduction depends on the hidden dimension $r$ but has little dependence on the input dimension $d$.

## 5.2 An Example of Functional Regularization via Masked Self-supervision

**Learning Without Functional Regularization.** Let $\mathcal{H}$ be linear functions from $\mathbb{R}^d$ to $\mathbb{R}^r$ where $r < (d-1)/2$ followed by a quadratic activation, and $\mathcal{F}$ be linear functions from $\mathbb{R}^r$ to $\mathbb{R}$. That is,

$$\phi = h_W(x) = [\sigma(w_1^\top x),\dots,\sigma(w_r^\top x)] \in \mathbb{R}^r , \; y = f_a(\phi) = a^\top \phi, \text{ where } w_i \in \mathbb{R}^d , a \in \mathbb{R}^r. \quad (9)$$

Here $\sigma(t)=t^2$ for $t \in \mathbb{R}$ is the quadratic activation function. W.l.o.g, we assume that $w_i$ and $a$ have $\ell_2$ norm bounded by 1. Without prior knowledge on the data, no functional regularization corresponds to end-to-end training on $\mathcal{F}\circ\mathcal{H}$.

**Data Property.** We consider the setting where the data point $x$ satisfies $x_1 = \sum_{i=1}^r ((u_i^*)^\top x_{2:d})^2$, where $x_{2:d} = [x_2, x_3, \dots, x_d]$ and $u_i^*$ is the $i$-th eigenvector of $\Sigma := \mathbb{E}[x_{2:d}x_{2:d}^\top]$. Furthermore, the label $y$ is given by $y = \sum_{i=1}^r a_i^*\sigma((u_i^*)^\top x_{2:d}) + \nu$ for some $\|a^*\|_2 \leq 1$ and a small Gaussian noise $\nu$. We also assume a difference in the $r^{\text{th}}$ and $r+1^{\text{th}}$ eigenvalues of $\Sigma$.

**Learning With Functional Regularization.** Suppose we have prior knowledge that $x_1 = \sum_{i=1}^r (u_i^\top x_{2:d})^2$ and $y = \sum_{i=1}^r a_i\sigma(u_i^\top x_{2:d})$ for some vectors $u_i \in \mathbb{R}^{d-1}$ and an $a$ with $\|a\|_2 \leq 1$. Based on this, we perform masked self-supervision by constraining the first coordinate of $w_i$ to be 0 for $h$, and choosing the regularization function $g(\phi)=\sum_{i=1}^r \phi_i$ and the regularization loss $L_r(h,g;x)=(x_1-g(h_W(x)))^2$. Again for simplicity, we assume access to infinite unlabeled data and set the regularization loss threshold $\tau = 0$.

On applying our framework, we get that functional regularization can reduce the labeled sample bound by $\frac{C}{\epsilon^2}\left[\ln\mathcal{N}_{\mathcal{H}}\left(\frac{\epsilon}{4L}\right) - \ln\mathcal{N}_{\mathcal{H}_{\mathcal{D}_X,L_r}(0)}\left(\frac{\epsilon}{4L}\right)\right]$ for some absolute constant $C$. We can derive the following using properties of $L_r$ and $g$ (Proof in Lemma 7 of Appendix C.2):

$$\mathcal{H}_{\mathcal{D}_X,L_r}(0)=\{h_W(x) : w_i=[0,u_i], [u_1,\dots,u_r]^\top=O[u_1^*,\dots,u_r^*]^\top, O \in \mathbb{R}^{r\times r}, O \text{ is orthonormal}\}$$

Using this we can show that the reduction of the sample bound is $\frac{C}{\epsilon^2}\ln\binom{d-1-r}{r}$, i.e., a reduction in the error bound by $\frac{C}{\sqrt{m_\ell}}\ln\binom{d-1-r}{r}$ when using $m_\ell$ labeled data. We also note that this reduction depends on $r$ but has little dependence on $d$.

# 6 Experiments

There is abundant empirical evidence on the benefits of auxiliary tasks in various applications, and hence we present experiments on verifying the benefits of functional regularization in Appendix E. Here we focus on experimentally verifying the following implications for the two concrete examples that we have analysed under our framework: 1) the reduction in prediction error (between end-to-end training and functional regularization) using the same amount of labeled data; 2) the reduction in prediction error on varying a property of the data and hypotheses (specifically, varying parameter $r$); 3) the pruning of the hypothesis class which results in reducing the prediction error. We present the complete experimental details in Appendix D for reproducibility, and additional results which verify that the reduction has little dependence on the input dimension $d$.

**Setup.** For the auto-encoder example, we randomly generate $d$ orthonormal vectors ($\{u_i^*\}_{i=1}^{i=d}$) in $\mathbb{R}^d$, means $\mu_i$ and variances $\sigma_i$ for $i \in [d]$ such that $\sigma_1 > \cdots > \sigma_r \gg \sigma_{r+1} > \cdots > \sigma_d$. We sample $\alpha_i \sim \mathcal{N}(\mu_i, \sigma_i) \; \forall i \in [d]$ and generate $x = \sum_{i=1}^d \alpha_i u_i$. For generating $y$, we use a randomly generated vector $a^* \in \mathbb{R}^r$. We use $d = 100$ and generate $10^4$ unlabeled, $10^4$ labeled training and $10^3$ labeled test points. We use the quadratic activation function and follow the specification in Section 5.1 (with regards to the hypothesis classes, reconstruction losses, etc.). For the masked self-supervision example, we similarly generate $x$ having the data property specified in Section 5.2 and then follow the other specifications in Section 5.2 (with regards to the hypothesis classes, reconstruction losses, etc.). We report the MSE on the test data points averaged over 10 runs as the metric.

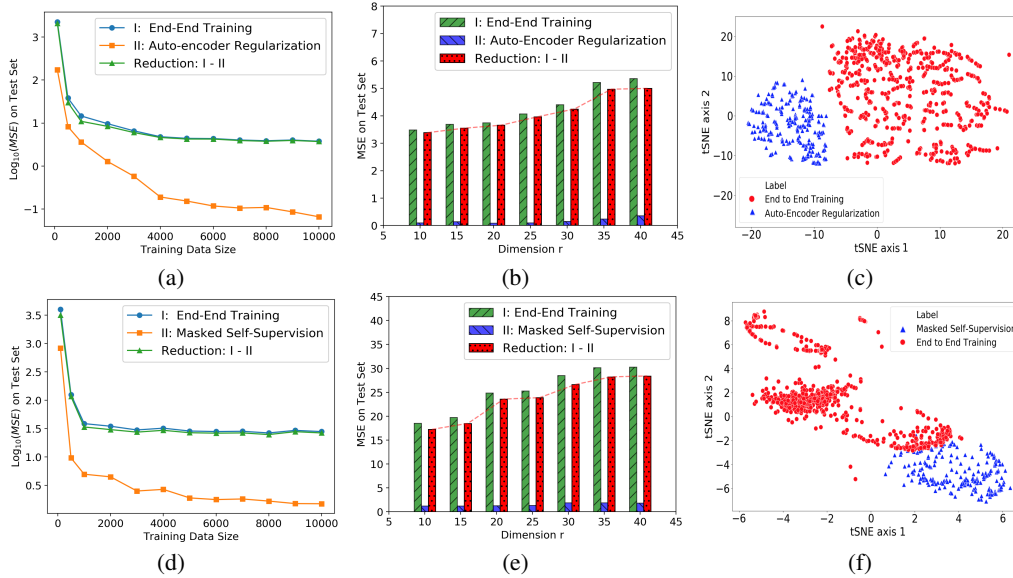

Figure 1: **Auto-Encoder:** 1(a) shows the variation of test MSE with labeled data size (here $r$=30), 1(b) shows this variation with the parameter $r$, and 1(c) shows the 2D visualization of the functional approximation using t-SNE. **Masked Self-Supervision:** 1(d), 1(e) and 1(f) show the same corresponding plots. Reduction refers to Test MSE of end-to-end training - Test MSE with regularization.

To support our key theoretical intuition that functional regularization prunes the hypothesis classes, we seek to visualize the learned model. Since multiple permutations of model parameters can result in the same model behavior, we visualise the function (from input to output) represented instead of the parameters using the method from [19]. Formally, we concatenate the outputs $f(h(x))$ on the test set points from a trained model into a single vector and visualise the vector in 2D using t-SNE [54]. We perform 1000 independent runs for each of the two models (with and without functional regularization) and plot the vectors for comparison. See Appendix D.1 for more details.

**Results.** Figure 1(a) plots the Test MSE loss by varying the size of the labeled data when $r = 30$. We observe that with the same labeled data size, functional regularization can significantly reduce the error compared to end-to-end training. Equivalently, it needs much fewer labeled samples to achieve the same error as end-to-end training (e.g., 500 v.s. 10,000 points). Also, the error without regularization does not decrease for sample sizes $\geq$ 2000 while it decreases with regularization, suggesting that the regularization can even help alleviate optimization difficulty. Figure 1(b) shows the effect of varying $r$ (i.e., the dimension of the subspace containing signals for prediction). We observe that the reduction in the error increases roughly linearly with $r$ and then grows slower, as predicted by our analysis. Figure 1(c) visualizes the prediction functions learned. It shows that when using the functional regularization, the learned functions stay in a small functional space, while they are scattered otherwise. This supports the intuition behind our theoretical analysis. This result also interestingly suggests that pruning the representation hypothesis space via functional regularization translates to a compact functional space for the prediction, even through optimization. We make similar observations for the masked self-supervision example in Figure 1(d)-1(f), which provide additional support for our analysis.

# 7 Conclusion

In this paper we have presented a unified discriminative framework for analyzing several representation learning approaches using unlabeled data, by viewing them as imposing a regularization on the representation via a learnable function. We have derived sample complexity bounds under various assumptions on the hypothesis classes and data, and shown that the functional regularization can be used to prune the hypothesis class and reduce the labeled sample complexity. We have also applied our framework to two concrete examples. An interesting future work direction is to investigate the effect of such functional regularization on the optimization of the learning methods.

## Broader Impact

Our paper is mostly theoretical in nature and thus we foresee no immediate negative societal impact. We are of the opinion that our theoretical framework may inspire development of improved representation learning methods on unlabeled data, which may have a positive impact in practice. In addition to the theoretical machine learning community, we perceive that our precise and easy-to-read formulation of unsupervised learning for downstream tasks can be highly beneficial to engineering-inclined machine learning researchers.

## Funding Disclosure

The work in this paper was funded in part by FA9550-18-1-0166 and IIS-2008559. The authors would also like to acknowledge the support provided by the University of Wisconsin-Madison Office of the Vice Chancellor for Research and Graduate Education with funding from the Wisconsin Alumni Research Foundation. No other funding entities with any competing interests influenced our work.

## Acknowledgements

The authors would like to thank the anonymous reviewers and the meta-reviewer for their valuable comments and suggestions which have been incorporated for the camera ready version.

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
