[Supplementary Material]

# Appendix

## A  Instantiations of Functional Regularization

Here we show that several unsupervised (self-supervised) representation learning strategies can be viewed as imposing a learnable function to regularize the representations being learned. We note that the class $\mathcal{G}$ can be an index set instead of a class of functions; our framework applies as long as the loss $L_r(h, g; x)$ is well defined (see the manifold learning example). $\mathcal{G}$ can also only have a single $g$, corresponding to the special case of a fixed regularizer (see the $\ell_p$ norm penalty example).

**Auto-encoder.**  Auto-encoders use an encoder function $h$ to map the input $x$ to a lower dimensional space $\phi$ and a decoder network $d$ to reconstruct the input back from $\phi$ using a MSE loss $\|x - d(h(x))\|^2$. One can view $d$ as a regularizer on the feature representation $\phi = h(x)$ through the regularization loss $L_r(h, g; x) = \|x - d(h(x))\|^2$. $\mathcal{H}_{\mathcal{D}_X, L_r}(\tau)$ is the subset of representation functions with at most $\tau$ reconstruction error using the best decoder in $\mathcal{G}$.

Variants of standard auto-encoders like sparse auto-encoders can be formulated similarly as a functional regularization on the representation being learnt.

**Masked Self-supervised Learning.**  Masked self-supervision techniques, in abstract terms, cover a portion of the input and then predict the masked input portion [13]. More concretely, say the input $x = [x_1, x_2, \ldots, x_d]$ is masked as $x' = [x_1, \ldots, x_i, 0, \ldots, 0, x_j, \ldots, x_d]$ and a function $g$ is learned to predict the masked input $[x_{i+1}, \ldots, x_{j-1}]$ over an input representation $h(x)$. This function $g$ used to reconstruct $x$, can be viewed as imposing a regularization on $h$ through a MSE regularization loss given by $\|x_{[i+1:j-1]} - g(h(x'))\|^2$. $\mathcal{H}_{\mathcal{D}_X, L_r}(\tau)$ is the subset of $\mathcal{H}$ which have at most $\tau$ MSE on predicting $x_{[i+1:j-1]}$ using the best function $g \in G$.

**Variational Auto-encoder.**  VAEs encode the input $x$ as a distribution $q_\phi(z|x)$ over a parametric latent space $z$ instead of a single point, and sample from it to reconstruct $x$ using a decoder $p_\theta(x|z)$. The encoder $q_\phi(z|x)$ is used to model the underlying mean $\mu_z$ and co-variance matrix $\sigma_z$ of the distribution over $z$. VAEs are trained by minimising a loss

$$\mathcal{L}_x(\theta, \phi) = -\mathbb{E}_{z \sim q_\phi(z|x)}[\log \ p_\theta(x|z)] + \mathbb{KL}(q_\phi(z|x)||p(z))$$

where $p(z)$ is specified as the prior distribution over $z$ (e.g., $\mathcal{N}(0, 1)$). The encoder $q_\phi(z|x)$ can be viewed as the representation function $h$, the decoder $p_\theta(x|z)$ as the learnable regularization function $g$, and the loss $\mathcal{L}_x(\theta, \phi)$ as the regularization loss $L_r(h, g; x)$ in our framework. Then $\mathcal{H}_{\mathcal{D}_X, L_r}(\tau)$ is the subset of encoders $q_\phi$ which have at most $\tau$ VAE loss when using the best decoder $p_\theta$ for it.

**Manifold Learning through the Triplet Loss.**  Learning manifold representations through metric learning is a popular technique used in computer vision applications [25]. A triplet loss formulation is used to learn a distance metric for the representations, by trying to minimise this metric between a baseline and positive sample and maximising the metric between the baseline and a negative sample. This is achieved by learning a representation function $h$ for an input $x$. Considering a triple of input samples $\bar{x} = (x_b, x_p, x_n)$ corresponding to a baseline, positive and negative sample, we use a loss $L_{\text{Triplet}}(\bar{x}) = \max(\|h(x_b) - h(x_p)\|_2^2 - \|h(x_b) - h(x_n)\|_2^2, 0)$ to learn $h$. This is a special instantiation of our framework using a dummy $\mathcal{G}$ having a single function $g$, where the regularization loss $L_r(h, g; \bar{x}) = L_{\text{Triplet}}(\bar{x})$ is computed over a triple of input samples.

Further, one can also consider some variants of the standard triplet loss formulation under our functional regularization perspective. For example, let the triplet loss be $L_{\text{Triplet}}^{(\alpha)}(\bar{x}) = \max(\|h(x_b) - h(x_p)\|_2^2 - \|h(x_b) - h(x_n)\|_2^2 + \alpha, 0)$ where $\alpha \in \mathbb{R}$ is a margin between the positive and negative pairs. When $\alpha$ is learnable, this corresponds to a functional regularization where $\mathcal{G} = \{\alpha : \alpha \in \mathbb{R}\}$, and the regularization loss is $L_r(h, g; \bar{x}) = L_{\text{Triplet}}^{(\alpha)}(\bar{x})$. In this case, the class $\mathcal{G}$ is not defined on top of the representation $h(x)$. However, our framework and the sample complexity analysis can still be applied through the definition of $L_r(h, g; \bar{x})$.

**Sparse Dictionary Learning.**  Sparse dictionary learning is an unsupervised learning approach to obtain a sparse low-dimensional representation of the input data. Here we consider a distributional

view of sparse dictionary learning. Give a distribution $\mathcal{D}_X$ over unlabeled data $x \in \mathbb{R}^d$ and a hyper-parameter $\lambda > 0$, we want to find a dictionary matrix $D \in \mathbb{R}^{d \times K}$ and a sparse representation $z \in \mathbb{R}^K$ for each $x$, so as to minimize the error $\mathbb{E}[L_D(x)]$, where $L_D(x)$ is the error on one point $x$ defined as $L_D(x) := \|x - Dz\|_2^2 + \lambda\|z\|_0$, subject to the constraint that each column of $D$ has $\ell_2$ norm bounded by 1. The learned representations $z$ can then be used for a target prediction task. Under our framework, we can view the representation function corresponding to $z = h_D(x) = \arg\min_{z \in \mathbb{R}^K} \|x - Dz\|_2^2 + \lambda\|z\|_0$, and $D$ is the parameter of the representation function. The regularization function class $\mathcal{G}$ has a single $g$, and the regularization loss is $L_r(h_D, g; x) = L_D(x)$.

Our framework also captures an interesting variant of dictionary learning. Consider another dictionary matrix $F$ and a hyper-parameter $\eta > 0$. The representation function still corresponds to $z = h_D(x) = \arg\min_{z \in \mathbb{R}^K} \|x - Dz\|_2^2 + \lambda\|z\|_0$, with $D$ as the parameter. The regularization function class is now given by $\mathcal{G} = \{g_F(z) = Fz : F \in \mathbb{R}^{d \times K}\}$, and the regularization loss $L_r(h_D, g_F; x)$ is defined as $\|x - g_F(h_D(x))\|_2^2 + \lambda\|z\|_0 + \eta\|D - F\|_F^2$. This special case of dictionary learning allows the encoding and decoding steps to use two different dictionaries $D$ and $E$ but constraining the difference between them. When $\eta \to +\infty$, this variant reduces to the original version described earlier.

**Explicit $\ell_p$ Norm Penalty.** Techniques imposing explicit regularizations on the representation $h$ being learned, often use an $\ell_p$ norm penalty on $h(x)$ i.e, $\|h(x)\|_p^p$ to the prediction loss while jointly training $f$ and $h$. This can be viewed as a special case of our framework using a fixed regularization function $g(h(x)) = \|h(x)\|_p^p$.

**Restricted Boltzmann Machines.** Restricted Boltzmann Machines (RBM) [48, 24] generate hidden representations for an input through unsupervised learning on unlabeled data. RBMs are characterized by a joint distribution over the input $x \in \{0,1\}^d$ and the representation $z \in \{0,1\}^r$: $P(x, z) = \frac{1}{Z}e^{-E(x,z)}$, where $Z$ is the partition function and $E(x, z)$ is the energy function defined as: $E(x, z) = -a^\top x - b^\top z - x^\top W z$, where $a \in \mathbb{R}^d, b \in \mathbb{R}^r, W \in \mathbb{R}^{d \times r}$ are parameters to be learned.

Then $P(z|x)$, for a fixed $x$, is a distribution parameterized by $b$ and $W$; which can be denoted as $q_{W,b}(z|x)$. Similarly, $P(x|z)$ is parameterized by $a$ and $W$ and thus can be denoted as $p_{W,a}(x|z)$. Given $x \sim \mathcal{D}_X$, the objective of the RBM is to minimize $-\mathbb{E}_{x \sim \mathcal{D}_X}[\log P(x)]$.

While the standard RBM objective does not have a direct analogy under our functional regularization framework, a heuristic variant can be formulated under our framework. If we use $\mathbb{E}_{P(x)}$ to denote the expectation over the marginal distribution of $x$ in the RBM, $\mathbb{E}_{P(z)}$ to denote the expectation over the marginal distribution of $z$, and $\mathbb{E}_{\mathcal{D}_X}$ to denote the expectation over $x \sim \mathcal{D}_X$. Then the following hold for the standard RBM:

$$P(z) = \mathbb{E}_{P(x)}[P(z|x)] = \mathbb{E}_{P(x)}[q_{W,b}(z|x)] \tag{10}$$

$$P(x) = \mathbb{E}_{P(z)}[P(x|z)] = \mathbb{E}_{P(z)}[p_{W,a}(x|z)] \tag{11}$$

In the heuristic variant, we replace $P(x)$ with $\mathcal{D}_X$ in Equation (10):

$$\hat{P}(z) = \mathbb{E}_{\mathcal{D}_X}[P(z|x)] = \mathbb{E}_{\mathcal{D}_X}[q_{W,b}(z|x)], \quad \hat{P}(x) = \mathbb{E}_{\hat{P}(z)}[P(x|z)] = \mathbb{E}_{\hat{P}(z)}[p_{W,a}(x|z)], \tag{12}$$

and train using the loss:

$$L(W, a, b; x) := -\log \hat{P}(x) = -\log \mathbb{E}_{\hat{P}(z)}\{p_{W,a}(x|z)\} = -\log \mathbb{E}_{\mathbb{E}_{\mathcal{D}_X}[q_{W,b}(z|x)]}\{p_{W,a}(x|z)\}. \tag{13}$$

Furthermore, on introducing another weight matrix $F \in \mathbb{R}^{d \times r}$ for $P(x|z)$ and a hyper-parameter $\eta > 0$, we can train the RBM using the loss:

$$L_\eta(W, a, b; x) := -\log \mathbb{E}_{\mathbb{E}_{\mathcal{D}_X}[q_{W,b}(z|x)]}\{p_{F,a}(x|z)\} + \eta\|W - F\|_F^2. \tag{14}$$

When $\eta \to +\infty$, this loss function reduces to the loss $L(W, a, b; x)$. Here $q_{W,b}(z|x)$ can be viewed as the representation function $h$ of our framework, $p_{F,a}(x|z)$ as the regularization function $g$, and $L_\eta(W, a, b; x)$ as the regularization loss $L_r(h, g; x)$.

**Comparison to GANs.** Finally, we would like to comment on Generative Adversarial Networks (GANs) [21]. While both functional regularization and GANs use auxiliary tasks having a function class, the goal of GANs is to learn a generative model using an auxiliary task through a discriminative function (the discriminator), while the goal of functional regularization is to learn a discriminative model using an auxiliary task which is usually (though not always) through a generative function (e.g., the decoder in auto-encoders).

## B Sample Complexity Bounds

### B.1 Same Domain, Realizable, Finite Hypothesis Classes

For simplicity, we begin with the realizable case, where the hypothesis classes contain functions $g^*, h^*, f^*$ with a zero prediction and regularization loss. Here we consider that the unlabeled $U$ and labeled $S$ samples are from the same domain distribution $\mathcal{D}_X$. We derive the following Theorem.

**Theorem 1.** *Suppose there exist $h^* \in \mathcal{H}, f^* \in \mathcal{F}, g^* \in \mathcal{G}$ such that $L_c(f^*, h^*; \mathcal{D}) = 0$ and $L_r(h^*, g^*; \mathcal{D}_X) = 0$. For any $\epsilon_0, \epsilon_1 \in (0, 1/2)$, a set $U$ of $m_u$ unlabeled examples and a set $S$ of $m_l$ labeled examples are sufficient to learn to an error $\epsilon_1$ with probability $1 - \delta$, where*

$$m_u \geq \frac{1}{\epsilon_0}\left[\ln|\mathcal{G}| + \ln|\mathcal{H}| + \ln\frac{2}{\delta}\right], \qquad m_l \geq \frac{1}{\epsilon_1}\left[\ln|\mathcal{F}| + \ln|\mathcal{H}_{\mathcal{D}_X, L_r}(\epsilon_0)| + \ln\frac{2}{\delta}\right]. \quad (5)$$

*In particular, with probability at least $1 - \delta$, all hypotheses $h \in \mathcal{H}, f \in \mathcal{F}$ with $L_c(f, h; S) = 0$ and $L_r(h; U) = 0$ will have $L_c(f, h; \mathcal{D}) \leq \epsilon_1$.*

*Proof.* We first show that with high probability, only the hypotheses $h$ in $\mathcal{H}_{\mathcal{D}_X, L_r}(\epsilon_0)$ have $L_r(h; U) = 0$. For a given pair $g$ and $h$ with $L_r(h, g; \mathcal{D}_X) \geq \epsilon_0$, the probability that $L_r(h, g; U) = 0$ is at most

$$\mathbb{P}[L_r(h, g; U) = 0] \leq (1 - \epsilon_0)^{m_u} \leq \frac{\delta}{2|\mathcal{H}||\mathcal{G}|} \quad (15)$$

for the given value of $m_u$. By the union bound, with probability at least $1 - \delta/2$, only those $g$ and $h$ with $L_r(h, g; \mathcal{D}_X) \leq \epsilon_0$ have $L_r(h, g; U) = 0$. Then only hypotheses $h \in \mathcal{H}_{\mathcal{D}_X, L_r}(\epsilon_0)$ have $L_r(h; U) = 0$.

Then we show that with high probability, for all $h \in \mathcal{H}_{\mathcal{D}_X, L_r}(\epsilon_0)$, only those $f$ and $h$ with $L_c(f, h; \mathcal{D}) \leq \epsilon_1$ can have $L_c(f, h; S) = 0$. Similarly as above, for a pair $f \in \mathcal{F}$ and $h \in \mathcal{H}_{\mathcal{D}_X, L_r}(\epsilon_0)$ with $L_c(f, h; \mathcal{D}) \geq \epsilon_1$, the probability that $L_c(f, h; S) = 0$ is at most

$$\mathbb{P}[L_c(f, h; S) = 0] \leq (1 - \epsilon_1)^{m_\ell} \leq \frac{\delta}{2|\mathcal{H}_{\mathcal{D}_X, L_r}(\epsilon_0)||\mathcal{G}|} \quad (16)$$

for the given value of $m_\ell$. By the union bound, with probability $1 - \delta/2$, for $f \in \mathcal{F}$ and $h \in \mathcal{H}_{\mathcal{D}_X, L_r}(\epsilon_0)$, only those with $L_c(f, h; \mathcal{D}) \leq \epsilon_1$ can have $L_c(f, h; S) = 0$, proving the theorem. $\square$

### B.2 Same Domain, Unrealizable Case, Infinite Hypothesis Classes

When the hypothesis classes are of an infinite size, we use metric entropy to measure the capacity. Suppose $\mathcal{H}$ is indexed by parameter set $\Theta_H$ with norm $\|\cdot\|_H$, $\mathcal{G}$ by $\Theta_G$ with norm $\|\cdot\|_G$, and $\mathcal{F}$ by $\Theta_F$ with norm $\|\cdot\|_F$. Assume that the losses are $L$-Lipschitz with respect to the parameters. That is,

$$|L_r(h_\theta, g; x) - L_r(h_{\theta'}, g; x)| \leq L\|\theta - \theta'\|_H, \forall g \in \mathcal{G}, x \in \mathcal{X},$$
$$|L_r(h, g_\theta; x) - L_r(h, g_{\theta'}; x)| \leq L\|\theta - \theta'\|_G, \forall h \in \mathcal{H}, x \in \mathcal{X},$$
$$|L_c(h_\theta, f; x) - L_c(h_{\theta'}, f; x)| \leq L\|\theta - \theta'\|_H, \forall f \in \mathcal{F}, x \in \mathcal{X},$$
$$|L_c(h, f_\theta; x) - L_c(h, f_{\theta'}; x)| \leq L\|\theta - \theta'\|_G, \forall h \in \mathcal{H}, x \in \mathcal{X}.$$

Let $\mathcal{N}_\mathcal{G}(\epsilon)$ be the $\epsilon$-covering number of $\mathcal{G}$ w.r.t. the associated norm. This is similarly defined for the other function classes.

The assumptions that the regularization and prediction losses are 0 are usually impractical due to noise in the data distribution. Realistically we may assume that there exist ground-truth functions that can make the regularization and prediction losses small. We begin by considering a setting where the prediction loss can be zero while the regularization loss is not.

**Theorem 4.** *Suppose there exist $h^* \in \mathcal{H}, f^* \in \mathcal{F}, g^* \in \mathcal{G}$ such that $L_c(f^*, h^*; \mathcal{D}) = 0$ and $L_r(h^*, g^*; \mathcal{D}_X) \leq \epsilon_r$. For any $\epsilon_0, \epsilon_1 \in (0, 1/2)$, a set $U$ of $m_u$ unlabeled examples and a set $S$ of $m_l$ labeled examples is sufficient to learn to an error $\epsilon_1$ with probability $1 - \delta$, where*

$$m_u \geq \frac{C}{\epsilon_0^2}\ln\frac{1}{\delta}\left[\ln\mathcal{N}_\mathcal{G}\left(\frac{\epsilon_0}{4L}\right) + \ln\mathcal{N}_\mathcal{H}\left(\frac{\epsilon_0}{4L}\right)\right], \quad (17)$$

$$m_l \geq \frac{C}{\epsilon_1}\ln\frac{1}{\delta}\left[\ln\mathcal{N}_\mathcal{F}\left(\frac{\epsilon_1}{4L}\right) + \ln\mathcal{N}_{\mathcal{H}_{\mathcal{D}_X, L_r}(\epsilon_r + \epsilon_0)}\left(\frac{\epsilon_1}{4L}\right)\right] \quad (18)$$

*for some absolute constant $C$. In particular, with probability at least $1 - \delta$, the hypotheses $f \in \mathcal{F}, h \in \mathcal{H}$ with $L_c(f, h; S) = 0$ and $L_r(h, g; U) \leq \epsilon_r + \epsilon_0$ for some $g \in \mathcal{G}$ satisfy $L_c(f, h; \mathcal{D}) \leq \epsilon_1$.*

*Proof.* First, we show that with $m_u$ unlabeled examples, by a covering argument over $\mathcal{H}$ and $\mathcal{G}$ (see, e.g., [55]), it is guaranteed that with probability $1 - \delta/2$, all $h \in \mathcal{H}$ and $g \in \mathcal{G}$ satisfy $|L_r(h, g; U) - L_r(h, g; \mathcal{D}_X)| \leq \epsilon_0$. More precisely, let $\mathcal{C}_{\mathcal{G}}\left(\frac{\epsilon_0}{4L}\right)$ be a $\frac{\epsilon_0}{4L}$-covering of $\mathcal{G}$, and $\mathcal{C}_{\mathcal{H}}\left(\frac{\epsilon_0}{4L}\right)$ be a $\frac{\epsilon_0}{4L}$-covering of $\mathcal{H}$. Then by the union bound, all $h' \in \mathcal{C}_{\mathcal{H}}\left(\frac{\epsilon_0}{4L}\right)$ and $g' \in \mathcal{C}_{\mathcal{G}}\left(\frac{\epsilon_0}{4L}\right)$ satisfy $|L_r(h', g'; U) - L_r(h', g'; \mathcal{D}_X)| \leq \epsilon_0/4$. Then the claim follows from the definition of the coverings and the Lipschitzness of the losses.

By the claim, we have $L_r(h^*, g^*; U) \leq L_r(h^*, g^*; \mathcal{D}_X) + \epsilon_0 \leq \epsilon_r + \epsilon_0$. So $h^* \in \mathcal{H}_{\mathcal{D}_X, L_r}(\epsilon_r + \epsilon_0)$, and thus the optimal value $L_c(f, h; S)$ subject to $L_r(h, g; U) \leq \epsilon_r + \epsilon_0$ for some $g \in \mathcal{G}$ is 0. On the other hand, again by a covering argument over $\mathcal{H}$ and $\mathcal{F}$, with probability at least $1 - \delta/2$, for all $h \in \mathcal{H}_{\mathcal{D}_X, L_r}(\epsilon_r + \epsilon_0)$ and all $f \in \mathcal{F}$, only those with $L_c(f, h; \mathcal{D}) \leq \epsilon_1$ can have $L_c(f, h; S) = 0$. The theorem statement then follows. □

The theorem shows that when the optimal regularization loss is not zero but $\epsilon_r > 0$, one needs to do the learning subject to $L_r(h; U) \leq \epsilon_r + \epsilon_0$ and the unlabeled sample complexity has a dependence on $\epsilon_0$ by $\frac{1}{\epsilon_0^2}$, instead of $\frac{1}{\epsilon_0}$.

We are now ready to present the result for the setting where both the optimal prediction and regularization losses are non-zero.

**Theorem 2.** *Suppose there exist $h^* \in \mathcal{H}, f^* \in \mathcal{F}, g^* \in \mathcal{G}$ such that $L_c(f^*, h^*; \mathcal{D}) \leq \epsilon_c$ and $L_r(h^*, g^*; \mathcal{D}_X) \leq \epsilon_r$. For any $\epsilon_0, \epsilon_1 \in (0, 1/2)$, a set $U$ of $m_u$ unlabeled examples and a set $S$ of $m_l$ labeled examples are sufficient to learn to an error $\epsilon_c + \epsilon_1$ with probability $1 - \delta$, where*

$$m_u \geq \frac{C}{\epsilon_0^2} \ln \frac{1}{\delta} \left[ \ln \mathcal{N}_{\mathcal{G}}\left(\frac{\epsilon_0}{4L}\right) + \ln \mathcal{N}_{\mathcal{H}}\left(\frac{\epsilon_0}{4L}\right) \right], \tag{6}$$

$$m_l \geq \frac{C}{\epsilon_1^2} \ln \frac{1}{\delta} \left[ \ln \mathcal{N}_{\mathcal{F}}\left(\frac{\epsilon_1}{4L}\right) + \ln \mathcal{N}_{\mathcal{H}_{\mathcal{D}_X, L_r}(\epsilon_r + 2\epsilon_0)}\left(\frac{\epsilon_1}{4L}\right) \right] \tag{7}$$

*for some absolute constant $C$. In particular, with probability at least $1 - \delta$, the $h \in \mathcal{H}, f \in \mathcal{F}$ that optimize $L_c(f, h; S)$ subject to $L_r(h; U) \leq \epsilon_r + \epsilon_0$ have $L_c(f, h; \mathcal{D}) \leq L_c(f^*, h^*; \mathcal{D}) + \epsilon_1$.*

*Proof.* With $m_u$ unlabeled examples, by a standard covering argument, it is guaranteed that with probability $1 - \delta/4$, all $h \in \mathcal{H}$ and $g \in \mathcal{G}$ satisfy $|L_r(h, g; U) - L_r(h, g; \mathcal{D}_X)| \leq \epsilon_0$. In particular, $L_r(h^*, g^*; U) \leq L_r(h^*, g^*; \mathcal{D}_X) + \epsilon_0 \leq \epsilon_r + \epsilon_0$. Then again by a covering argument, the labeled sample size $m_l$ implies that with probability at least $1 - \delta/2$, all hypotheses $h \in \mathcal{H}_{\mathcal{D}_X, L_r}(\epsilon_r + 2\epsilon_0)$ and all $f \in \mathcal{F}$ have $L_c(f, h; S) \leq L_c(f, h; \mathcal{D}) + \epsilon_1/2$. Finally, by using Hoeffding's bounds, with probability at least $1 - \delta/4$, we have

$$L_c(f^*, h^*; S) \leq L_c(f^*, h^*; \mathcal{D}) + \mathcal{O}\left(\sqrt{\frac{1}{m_l} \ln \frac{1}{\delta}}\right) \leq L_c(f^*, h^*; \mathcal{D}) + \epsilon_1/2.$$

Therefore, with a probability of at least $1 - \delta$, the hypotheses $f \in \mathcal{F}, h \in \mathcal{H}$ that optimizes $L_c(f, h; S)$ subject to $L_r(h, g; U) \leq \epsilon_r + \epsilon_0$ for some $g \in \mathcal{G}$ have the following guarantee. First, since $L_r(h, g; U) \leq \epsilon_r + \epsilon_0$, we have $L_r(h, g; \mathcal{D}_X) \leq \epsilon_r + 2\epsilon_0$, and thus $h \in \mathcal{H}_{\mathcal{D}_X, L_r}(\epsilon_r + 2\epsilon_0)$. Then we have

$$L_c(f, h; \mathcal{D}) \leq L_c(f, h; S) + \epsilon_1/2 \tag{19}$$
$$\leq L_c(f^*, h^*; S) + \epsilon_1/2 \tag{20}$$

$$\leq L_c(f^*, h^*; \mathcal{D}) + \mathcal{O}\left(\sqrt{\frac{1}{m_l} \ln \frac{1}{\delta}}\right) + \epsilon_1/2 \tag{21}$$

$$\leq L_c(f^*, h^*; \mathcal{D}) + \epsilon_1. \tag{22}$$

This completes the proof of the theorem. □

The above analysis also holds with some other capacity measure of the hypothesis classes, like the VC-dimension or Rademacher complexity. We give an example for using the VC-dimension (assuming the prediction task is a classification task). The proof follows similarly to Theorem 2, but using the VC-dimension bound instead of the $\epsilon$-net argument.

**Theorem 5.** *Suppose there exist $h^* \in \mathcal{H}, f^* \in \mathcal{F}, g^* \in \mathcal{G}$ such that $L_c(f^*, h^*; \mathcal{D}) \leq \epsilon_c$ and $L_r(h^*, g^*; \mathcal{D}_X) \leq \epsilon_r$. For any $\epsilon_0, \epsilon_1 \in (0, 1/2)$, a set $U$ of $m_u$ unlabeled examples and a set $S$ of $m_l$ labeled examples are sufficient to learn to an error $\epsilon_c + \epsilon_1$ with probability $1 - \delta$, where*

$$m_u \geq \frac{C}{\epsilon_0^2} \left[ d(\mathcal{G} \circ \mathcal{H}) \ln \frac{1}{\epsilon_0} + \ln \frac{1}{\delta} \right], \quad m_l \geq \frac{C}{\epsilon_1^2} \left[ d(\mathcal{F} \circ \mathcal{H}_{\mathcal{D}_X, L_r}(\epsilon_r + 2\epsilon_0)) \ln \frac{1}{\epsilon_1} + \ln \frac{1}{\delta} \right] \quad (23)$$

*for some absolute constant $C$. In particular, with probability at least $1 - \delta$, the hypotheses $f \in \mathcal{F}, h \in \mathcal{H}$ that optimize $L_c(f, h; S)$ subject to $L_r(h; U) \leq \epsilon_r + \epsilon_0$ satisfy $L_c(f, h; \mathcal{D}) \leq L_c(f^*, h^*; \mathcal{D}) + \epsilon_1$.*

### B.3 Different Domains, Unrealizable, Infinite Hypothesis Classes

In practice, it is often the case that the unlabeled data is from a different domain than the labeled data. For example, state-of-the-art NLP systems are often trained on a large general unlabeled corpus (e.g., the entire Wikipedia) and a small specific labeled corpus (e.g., a set of medical records). That is, the unlabeled data $U$ is from a distribution $\mathcal{U}_X$ different from $\mathcal{D}_X$, the marginal distribution of $x$ in the labeled data. In this setting, we show that our previous analysis still holds.

**Theorem 3.** *Suppose the unlabeled data $U$ is from a distribution $\mathcal{U}_X$ different from $\mathcal{D}_X$. Suppose there exist $h^* \in \mathcal{H}, f^* \in \mathcal{F}, g^* \in \mathcal{G}$ such that $L_c(f^*, h^*; \mathcal{D}) \leq \epsilon_c$ and $L_r(h^*, g^*; \mathcal{U}_X) \leq \epsilon_r$. Then the same sample complexity bounds as in Theorem 2 hold (replacing $\mathcal{D}_X$ with $\mathcal{U}_X$ in Equation 7).*

*Proof.* The proof follows that for the setting with the same distribution for input feature vectors in the labeled data and unlabeled data; here we only mention the proof steps involving $\mathcal{U}_X$.

Even when the unlabeled data is from a different distribution $\mathcal{U}_X$, we still have that with probability $1 - \delta/4$, all $h \in \mathcal{H}$ and $g \in \mathcal{G}$ satisfy $|L_r(h, g; U) - L_r(h, g; \mathcal{U}_X)| \leq \epsilon_0$ for the given value of $m_u$. In particular, $L_r(h^*, g^*; U) \leq L_r(h^*, g^*; \mathcal{U}_X) + \epsilon_0 \leq \epsilon_r + \epsilon_0$. Then the labeled sample size $m_l$ implies that with probability at least $1 - \delta/2$, all hypotheses $h \in \mathcal{H}_{\mathcal{U}_X, L_r}(\epsilon_r + 2\epsilon_0)$ and all $f \in \mathcal{F}$ have $L_c(f, h; S) \leq L_c(f, h; \mathcal{D}) + \epsilon_1/2$. Also, for any $h, g$ with $L_r(h, g; U) \leq \epsilon_r + \epsilon_0$, we have $L_r(h, g; \mathcal{U}_X) \leq \epsilon_r + 2\epsilon_0$, and thus $h \in \mathcal{H}_{\mathcal{D}_X, L_r}(\epsilon_r + 2\epsilon_0)$. The rest of the proof follows that of Theorem 2. □

**Remarks** We would like to briefly comment on interpreting the reduction in sample complexity of labeled data when using functional regularization in our bounds. The sample complexity bounds are *upper bounds* and are aimed at aiding quantitative analysis by bounding the actual sample size needed for learning (under assumptions on the data and the hypothesis class). However, there exist settings where these bounds are nearly tighter mathematically (e.g., the standard lower bound via VC-dimension). More precisely, there exist hypothesis classes, such that for any learning algorithm, there exists a data distribution and a target function such that a sample, equal in size to the upper bound up to logarithmic factors, is required for learning (a more precise statement can be found in [36]). Additionally, these bounds usually do not take into account the effect of optimization [59].

While these upper bounds are not an exact quantification, they usually align well with the sample size needed for learning in practice, thereby providing useful insights. The reduction in our bounds on using functional regularization can roughly estimate the actual reduction in practice. Further this can provide useful theoretical insights such as the regularization restricting the learning to a subset of the hypothesis class of representation functions. Similar to prior sample complexity studies, we believe our sample complexity bounds can prove to be a useful analysis tool.

## C Proofs for Applying the Theoretical Framework to Concrete Examples

### C.1 Auto-encoder

We first recall the details of the example: $\mathcal{H}$ is the class of linear functions from $\mathbb{R}^d$ to $\mathbb{R}^r$ where $r < d/2$, and $\mathcal{F}$ to be the class of linear functions over some activations. That is,

$$z = h_W(x) = Wx, \quad y = f_a(z) = \sum_{i=1}^{r} a_i \sigma(z_i), \text{ where } W \in \mathbb{R}^{r \times d}, \ a \in \mathbb{R}^r \quad (8)$$

Here $\sigma(t)$ is an activation function, the rows of $W$ and $a$ have $\ell_2$ norm bounded by 1. We consider the Mean Square Error (MSE) prediction loss, i.e., $L_c(f, h; x) = \|y - f(h(x))\|_2^2$.

Also recall that we assume the data distribution having the following property: let the columns of $B \in \mathbb{R}^{d \times d}$ be the eigenvectors of $\Sigma := \mathbb{E}[xx^\top]$, then the labels are largely determined by the signal in the first $r$ directions: $y = (a^*)^\top z^* + \nu$ and $z^* = B_{1:r}^\top x$, where $a^*$ is a ground-truth parameter with $\|a^*\|_2 \leq 1$, $B_{1:r}$ is the set of first $r$ eigenvectors of $\Sigma$, and $\nu$ is a small Gaussian noise. We also assume that the $r^{\text{th}}$ and $r+1^{\text{th}}$ eigenvalues of $\Sigma$ are different so that the corresponding eigenvectors can be distinguished. Let $\epsilon_r$ denote $\mathbb{E}\|x - B_{1:r}B_{1:r}^\top x\|_2^2$.

Finally, we recall that the functional regularization $\mathcal{G}$ we used is given by the class of linear functions from $\mathbb{R}^r$ to $\mathbb{R}^d$, i.e., $\hat{x} = g_V(z) = Vz$ where $V \in \mathcal{R}^{d \times r}$ with orthonormal columns. The regularization loss $L_r(h, g; x) = \|x - g(h(x))\|_2^2$.

For simplicity of analysis, we assume access to infinite unlabeled data, and set the threshold $\tau = \epsilon_r$. Strictly speaking, we need to allow $L_r(h, g; \mathcal{D}_X) \leq \epsilon_r + \epsilon$ for a small $\epsilon > 0$ due to finite unlabeled data. A similar but more complex argument holds for that case. Here we assume infinite unlabeled data to simplify the presentation and better illustrate the intuition, since our focus is on quantifying the reduction in labeled data.

Formally, we calculate the sample complexity bounds in the limit $m_u \to +\infty$. Equivalently we consider the learning problem:

$$\min_{f \in \mathcal{F}, h \in \mathcal{H}} L_c(f, h\,; S), \quad \text{s.t. } L_r(h\,; \mathcal{D}_X) \leq \epsilon_r. \tag{24}$$

Let $\mathcal{N}_{\mathcal{C}}(\epsilon)$ denote the $\epsilon$-covering number of a class $\mathcal{C}$ w.r.t. the $\ell_2$ norm (i.e., Euclidean norm for the weight vector $a$, and Frobenius norm for the weight matrices $W$ and $V$). Let $L$ denote the Lipschitz constant of the losses (See Appendix B.2). Without regularization, the standard $\epsilon$-net argument shows that the labeled sample complexity, for an error $\epsilon$ close to the optimal, is $\frac{C}{\epsilon^2}\left[\ln \mathcal{N}_{\mathcal{F}}\left(\frac{\epsilon}{4L}\right) + \ln \mathcal{N}_{\mathcal{H}}\left(\frac{\epsilon}{4L}\right)\right]$ for some absolute constant $C$. Applying our framework when using regularization, the sample complexity is bounded by $\frac{C}{\epsilon^2}\left[\ln \mathcal{N}_{\mathcal{F}}\left(\frac{\epsilon}{4L}\right) + \ln \mathcal{N}_{\mathcal{H}_{\mathcal{D}_X, L_r}(\epsilon_r)}\left(\frac{\epsilon}{4L}\right)\right]$. To quantify the reduction in the bound, we show the following lemma.

**Lemma 6.** *For $\epsilon/4L < 1/2$,*

$$\mathcal{N}_{\mathcal{H}}\left(\frac{\epsilon}{4L}\right) \geq \binom{d-r}{r} \mathcal{N}_{\mathcal{H}_{\mathcal{D}_X, L_r}(\epsilon_r)}\left(\frac{\epsilon}{4L}\right). \tag{25}$$

*Proof.* First, recall that the regularization loss is

$$
\begin{aligned}
L_r(h, g; \mathcal{D}_X) &= \mathbb{E}_x \|x - g(h(x))\|_2^2 \\
&= \mathbb{E}_x \|x - VWx\|_2^2
\end{aligned} \tag{26}
$$

which is the $r$-rank approximation of the data. So in the optimal solution, the columns of $V$ and the rows of $W$ should span the subspace of the top $r$ eigenvectors $\Sigma$. More precisely,

$$
\begin{aligned}
L_r(h, g; \mathcal{D}_X) &= \mathbb{E}_x[x^\top (I - VW)^\top (I - VW)x] \\
&= \mathbb{E}_x[\text{trace}(x^\top (I - VW)^\top (I - VW)x)] \\
&= \mathbb{E}_x[\text{trace}((I - VW)^\top (I - VW)xx^\top)] \\
&= \text{trace}((I - VW)^\top (I - VW)\Sigma). \\
&= \text{trace}((I - VW)\Sigma). 
\end{aligned} \tag{27}
$$

Since $V$ and $W$ are orthonormal and have rank $r$, the optimal $VW$ should span the subspace of the top $r$ eigenvectors of $\Sigma$ and the optimal loss is given by $\epsilon_r$.[2] Since the $r$-th and $r+1$-th eigenvalues of $\Sigma$ are different, the optimal $VW$ is unique, and thus we have

$$\mathcal{H}_{\mathcal{D}_X, L_r}(\epsilon_r) = \{OB_{1:r}^\top : O \in \mathbb{R}^{r \times r}, O \text{ is orthonormal}\}.$$

On the other hand, if $B_S$ refers to the sub-matrix of columns in $B$ having indices in $S$, then clearly,

$$\mathcal{H} \supseteq \mathcal{H}_S := \{OB_S^\top : O \in \mathbb{R}^{r \times r}, O \text{ is orthonormal}\},$$

for any $S \subseteq \{r+1, r+2, \ldots, d\}, |S| = r$. By symmetry, $\mathcal{N}_{\mathcal{H}_S}(\epsilon') = \mathcal{N}_{\mathcal{H}_{\mathcal{D}_X, L_r}(\epsilon_r)}(\epsilon')$ for any $\epsilon' > 0$. Now it is sufficient to prove that $\mathcal{H}_S$ and $\mathcal{H}_{S'}$ are sufficiently far away for different $S$ and $S'$. This is indeed the case, since $\|OB_S^\top - O'B_{S'}^\top\|_F^2 > 1$ for any orthonormal $O$ and $O'$:

$$\|OB_S^\top - O'B_{S'}^\top\|_F^2 = \operatorname{trace}\left((OB_S^\top - O'B_{S'}^\top)^\top(OB_S^\top - O'B_{S'}^\top)\right) \tag{28}$$

$$= \operatorname{trace}\left((OB_S^\top)^\top(OB_S^\top)\right) + \operatorname{trace}\left((O'B_{S'}^\top)^\top(O'B_{S'}^\top)\right)$$
$$- \operatorname{trace}\left((OB_S^\top)^\top(O'B_{S'}^\top)\right) - \operatorname{trace}\left((O'B_{S'}^\top)^\top(OB_S^\top)\right) \tag{29}$$

$$= \|OB_S^\top\|_F^2 + \|O'B_{S'}^\top\|_F^2$$
$$- \operatorname{trace}\left((O'B_{S'}^\top)(OB_S^\top)^\top\right) - \operatorname{trace}\left((OB_S^\top)(O'B_{S'}^\top)^\top\right) \tag{30}$$

$$= \|B_S^\top\|_F^2 + \|B_{S'}\|_F^2 - \operatorname{trace}\left(B_{S'}^\top B_S\right) - \operatorname{trace}\left(B_S^\top B_{S'}\right) \tag{31}$$

$$\geq r + r - (r-1) - (r-1) = 2. \tag{32}$$

This completes the proof. $\qquad\square$

### C.2 Masked Self-supervision

We first recall the details of the example: $\mathcal{H}$ is the class of linear functions from $\mathbb{R}^d$ to $\mathbb{R}^r$ where $r < (d-1)/2$ followed by a quadratic activation, and $\mathcal{F}$ is the class of linear functions from $\mathbb{R}^r$ to $\mathbb{R}$. That is,

$$z = h_W(x) = [\sigma(w_1^\top x), \ldots, \sigma(w_r^\top x)] \in \mathbb{R}^r \ , \ y = f_a(z) = a^\top z, \text{ where } w_i \in \mathbb{R}^d, a \in \mathbb{R}^r. \tag{9}$$

Here $\sigma(t) = t^2$ for $t \in \mathbb{R}$ is the quadratic activation function. Since $\sigma(ct) = c^2 t^2$, by scaling, w.l.o.g. we can assume that $\|w_i\|_2 = 1$ and $\|a\|_2 \leq 1$. Without prior knowledge of data, no regularization refers to end-to-end training on $\mathcal{F} \circ \mathcal{H}$.

Also recall that we assume the data $x$ satisfies $x_1 = \sum_{i=1}^r ((u_i^*)^\top x_{2:d})^2$, where $x_{2:d} = [x_2, x_3, \ldots, x_d]$ and $u_i^*$ is the $i$-th eigenvector of $\Sigma := \mathbb{E}[x_{2:d} x_{2:d}^\top]$. Furthermore, the label $y$ is given by $y = \sum_{i=1}^r a_i^* \sigma((u_i^*)^\top x_{2:d}) + \nu$ for some $\|a^*\|_2 \leq 1$ and a small Gaussian noise $\nu$. We also assume a significant difference in the $r^{\text{th}}$ and $r+1^{\text{th}}$ eigenvalues of $\Sigma$.

Finally, we recall that we used a masked self-supervision functional regularization by constraining the first coordinate of $w_i$ to be 0 for $h$, and choosing the regularization function $g(z) = \sum_{i=1}^r z_i$ and the regularization loss $L_r(h, g; x) = (x_1 - g(h_W(x)))^2$. Note that there is only one $g \in \mathcal{G}$, which is a special case of our framework and our sample complexity theorems still apply.

Again for simplicity, we assume access to infinite unlabeled data, and set the threshold $\tau = 0$. Our framework shows that the functional regularization via masked self-supervision reduces the labeled sample bound by $\frac{C}{\epsilon^2}\left[\ln \mathcal{N}_{\mathcal{H}}\left(\frac{\epsilon}{4L}\right) - \ln \mathcal{N}_{\mathcal{H}_{\mathcal{D}_X, L_r}(0)}\left(\frac{\epsilon}{4L}\right)\right]$ for some absolute constant $C$. The following lemma then gives an estimation of this reduction.

**Lemma 7.** *For $\epsilon/4L < 1/2$,*

$$\mathcal{N}_{\mathcal{H}}\left(\frac{\epsilon}{4L}\right) \geq \binom{d-1-r}{r} \mathcal{N}_{\mathcal{H}_{\mathcal{D}_X, L_r}(0)}\left(\frac{\epsilon}{4L}\right). \tag{33}$$

*Proof.* By definition,

$$\mathbb{E}\left[L_r(h, g; x)\right] = \mathbb{E}\left[\sum_{i=1}^r u_i^\top x_{2:d} x_{2:d}^\top u_i - \sum_{i=1}^r (u_i^*)^\top x_{2:d} x_{2:d}^\top u_i^*\right]^2 \tag{34}$$

$$\geq \left(\mathbb{E}\left|\sum_{i=1}^r u_i^\top x_{2:d} x_{2:d}^\top u_i - \sum_{i=1}^r (u_i^*)^\top x_{2:d} x_{2:d}^\top u_i^*\right|\right)^2 \tag{35}$$

$$\geq \left|\mathbb{E}\sum_{i=1}^r u_i^\top x_{2:d} x_{2:d}^\top u_i - \mathbb{E}\sum_{i=1}^r (u_i^*)^\top x_{2:d} x_{2:d}^\top u_i^*\right|^2, \tag{36}$$

$$\geq \left|\sum_{i=1}^r u_i^\top \Sigma u_i - \sum_{i=1}^r (u_i^*)^\top \Sigma u_i^*\right|^2. \tag{37}$$

Therefore, $\mathbb{E}\left[L_r(h, g; x)\right] = 0$ if and only if $u_1, \ldots, u_r$ span the same subspace as $u_1^*, \ldots, u_r^*$, i.e.,

$$\mathcal{H}_{\mathcal{D}_X, L_r}(0) = \{h_W(x) : w_i = [0, u_i], [u_1, \ldots, u_r]^\top = O[u_1^*, \ldots, u_r^*]^\top, O \in \mathbb{R}^{r \times r}, O \text{ is orthonormal}\}.$$

On the other hand, if $u_1^*, u_2^*, \ldots, u_{d-1}^*$ are the eigenvectors of $\Sigma$, and $U_I^* := [u_{i_1}^*, \ldots, u_{i_r}^*]^\top$ for indices $I = \{i_1, i_2, \ldots, i_r\} \subseteq \{r+1, r+2, \ldots, d-1\}$, then clearly

$$\mathcal{H} \subseteq \mathcal{H}_I := \{h_W(x) : w_i = [0, u_i], [u_1, \ldots, u_r]^\top = O(U_I^*)^\top, O \in \mathbb{R}^{r \times r}, O \text{ is orthonormal}\}$$

for any $I = \{i_1, i_2, \ldots, i_r\} \subseteq \{r+1, r+2, \ldots, d-1\}$. By symmetry, $\mathcal{N}_{\mathcal{H}_I}(\epsilon') = \mathcal{N}_{\mathcal{H}_{\mathcal{D}_X, L_r}(0)}(\epsilon')$ for any $\epsilon' > 0$. Using an argument similar to Section C.1, we can show that for two different index sets $I$ and $I'$, any hypothesis in $\mathcal{H}_I$ and any hypothesis in $\mathcal{H}_{I'}$ cannot be covered by the same ball in any $\epsilon'$-cover with $\epsilon' < 1/2$. This completes the proof. $\square$

# D  Experiments on Concrete Functional Regularization Examples

## D.1  Auto-Encoder

**Data:**  We first generate $d$ orthonormal vectors($\{u_i\}_{i=1}^{i=d}$) in $\mathbb{R}^d$. We then randomly generate means $\mu_i$ and variances $\sigma_i$ corresponding to each principal component $i \in [1, d]$ such that $\sigma_1 > \cdots > \sigma_r \gg \sigma_{r+1} > \cdots > \sigma_d$. The $\mu_i$'s are randomly generated integers in $[0, 20]$ and the variances $\sigma_i$, $i \in [1, r]$ are each generated randomly from $[1, 10]$ and $\sigma_i$, $i \in [r+1, d]$ are each generated randomly from $[0, 0.1]$. We also generate a vector $a \in \mathbb{R}^r$ randomly such that $||a||_2 \leq 1$. To generate a data point $(x, y)$, we sample $\alpha_i \sim \mathcal{N}(\mu_i, \sigma_i)\ \forall i \in [1, d]$ and set $x = \sum_{i=1}^d \alpha_i u_i$ and $y = \sum_{i=1}^r a_i \alpha_i^2 + \nu$ where $\nu \sim \mathcal{N}(0, 10^{-2})$. We use an unlabeled dataset of $10^4$ points (when using the auto-encoder functional regularization), a labeled training set of $10^4$ points and a labeled test set of $10^3$ points.

**Models:**  $h_W$ corresponds to a fully connected NN, without any activation function, to transform $x \in \mathbb{R}^d$ to its representation $h(s) \in \mathbb{R}^r$. For prediction on the target task, we use a linear classifier after a quadratic activation on $h(x)$ to obtain a scalar output $\hat{y}$. For functional regularization $g_V$, we use a fully connected NN to transform the representation $h(s) \in \mathbb{R}^r$ to reconstruct the input back $\hat{x} \in \mathbb{R}^d$. Our example additionally constrains $V, W$ to be orthonormal. For achieving this, we add an orthonormal regularization [10, 5] penalty for each $V, W$ weighted by hyper-parameters $\lambda_1$ and $\lambda_2$ respectively during the auto-encoder reconstruction. For a matrix $M \in R^{a \times b}$, the orthonormal regularization penalty to ensure that the rows of $M$ are orthonormal, is given by $\sum_{ij} |(MM^\top)_{ij} - I_{ij}|$ where $\sum_{ij}$ is summing over all the matrix elements, $I$ is the identity matrix in $R^{a \times a}$.

**Training Details:**  For end-to-end training, we train the predictor and $h$ jointly using a MSE loss between the predicted target $\hat{y}$ and the true $y$ on the labeled training data set. For functional regularization, we first train $h$ and $g$ using the MSE loss between the reconstructed input $\hat{x}$ and the original input $x$ over the unlabeled data set. Here, we also add the orthonormal regularization penalties. We tune the weights $\lambda_1$ and $\lambda_2$ using grid search in $[10^{-3}, 10^3]$ in multiplicative steps of 10 to get the best reconstruction (least MSE) on the training data inputs. Now using $h$ initialized from the auto-encoder, we use the labeled training data set to jointly learn the predictor and $h$ using a MSE loss between the predicted target $\hat{y}$ and the true $y$. We report the MSE on the test set as the metric. For all optimization steps we use an SGD optimizer with momentum set to 0.9 where the learning rate is tuned using grid search in $[10^{-5}, 10^{-1}]$ in multiplicative steps of 10. We set the data dimension $d = 100$ and report the test MSE averaged over 10 runs.

**t-SNE Plots of Functional Approximations**  To get a functional approximation from a model, we compute and concatenate the output predictions $\hat{y}$ from the model over the test data set. For every model, we obtain a $\mathbb{R}^{1000}$ vector corresponding to the size of the test set. We perform 1000 independent runs for each model (with and without functional regularization) obtaining $2,000$ functional approximation vectors in $\mathbb{R}^{1000}$. We visualise these vectors in 2D using the t-SNE [54] algorithm.

**Varying Dimension $d$:**  We plot the reduction in test MSE between end-to-end training and using functional regularization on varying $d$ in Figure 2. Here we fix $r = 30$ and vary the data dimension $d$ and present the test MSE scores normalized with the average norm $||x||_2^2$ over the test data. As per indications from our derived bounds, the reduction remains more or less constant on varying $d$.

(a) Auto-Encoder         (b) Masked Self-Supervision

Figure 2: Reduction in Test MSE (on using functional regularization with respect to end-to-end training) with dimension $d$. Here $r = 30$ and the Test MSE are normalized by the average test $\|x\|_2^2$.

## D.2    Masked Self-Supervision

**Data:** We first generate $d-1$ orthonormal vectors($\{u_i\}_{i=2}^{i=d}$) in $\mathbb{R}^{d-1}$. We then randomly generate means $\mu_i$ and variances $\sigma_i$ corresponding to each principal component $i \in [2, d]$ such that $\sigma_2 > \cdots > \sigma_{r+1} \gg \sigma_{r+2} > \cdots > \sigma_d$. The $\mu_i$'s are randomly generated integers in $[0, 20]$ and the variances $\sigma_i$, $i \in [2, r+1]$ are each generated randomly from $[1, 10]$ and $\sigma_i$, $i \in [r+2, d]$ are each generated randomly from $[0, 0.1]$. We also generate a vector $a \in \mathbb{R}^r$ randomly such that $||a||_2 \leq 1$. To generate a data point $(x, y)$, we sample $\alpha_i \sim \mathcal{N}(\mu_i, \sigma_i)$ $\forall i \in [2, d]$ and set $x_1 = \sum_{i=2}^{r+1} \alpha_i^2$, $x_{2:d} = \sum_{i=2}^{d} \alpha_i u_i$ and $y = \sum_{i=2}^{r+1} a_i \alpha_i^2 + \nu$ where $\nu \sim \mathcal{N}(0, 10^{-2})$. We use an unlabeled dataset of $10^4$ points, a labeled training set of $10^4$ points and a labeled test set of $10^3$ points.

**Models:** $h_W$ corresponds to a fully connected NN, using a quadratic activation function, to transform $x \in \mathbb{R}^d$ to its representation $h(s) \in \mathbb{R}^r$. For prediction on the target task we use a linear classifier to obtain the output $\hat{y}$ from the representation $h(x)$. For functional regularization, we sum the elements of the representation $h(x) \in \mathbb{R}^r$ to reconstruct the first input $\hat{x}_1 \in \mathbb{R}$ back.

**Training Details:** For functional regularization, we mask the first dimension of the unlabeled data by setting it to 0 and train $h$ using the MSE loss between the reconstructed $\hat{x}_1$ and the original input dimension $x_1$. Other experimental details remain similar to Section D.1.

Experimental details for the t-SNE plots of functional approximation remain similar to Section D.1.

**Varying Dimension** $d$**:** Following similar details to Section D.1, we present the graph in Figure 2. We can observe that the reduction does not change much on varying $d$ here as well.

# E    Additional Experiments on Functional Regularization

There have been several empirical studies verifying the benefits of functional regularization across different applications. Here we present empirical results showing the benefits of using functional regularization on a computer vision and natural language processing application.

## E.1    Image Classification

We consider the application of image classification using the Fashion MNIST dataset [57] which contains $28 \times 28$ gray-scale images of fashion products from 10 categories. This dataset has 60k images for training and 10k images for testing. We consider a denoising auto-encoder functional regularization using unlabeled data and evaluate its benefits to supervised classification using labeled data.

**Experimental Details** We use a denoising auto-encoder as the functional regularization when learning from unlabeled data. The encoder consists of three fully connected layers with ReLU activations to obtain the input representation $h(x)$ of 1024 dimensions from an input $x$. The decoder consists of three fully connected layers with ReLU activations to reconstruct the $28 \times 28$ image $\hat{x}$ back from the 1024 dimensional representation $h(x)$. For training, the pixel values of $x$ are normalized to

Figure 3: Experimental results on **Fashion-MNIST**. (a) Test accuracy using de-noising auto-encoder functional regularization compared to end-to-end training on varying the size of labeled training data. (b) The 2D visualization of the functional approximation of 100 independent runs for each method.

$[0, 1]$ and independently corrupted by adding a Gaussian noise with mean $0$ and standard deviation $0.2$. The MSE loss between the $x$ and $\hat{x}$ is used as the regularization loss $L_r$. Training is performed using the Adam optimizer with a learning rate of $3 \times 10^{-4}$. For classification, we use a simple linear layer which maps $h(x)$ to the class label $\hat{y}$. The classifier and the encoder are trained jointly using the cross entropy loss between $\hat{y}$ and the original label $y$. We compare the test set accuracy of 1) directly training the encoder and the target classifier using the labeled training data, and 2) pre-training the encoder using the de-noising auto-encoder functional regularization and then fine-tuning its weights along with the target classifier using the labeled training data. We vary the size of the labeled training data and plot the test accuracy averaged across 5 runs in Figure 3(a).

To visualize the impact of the denoising auto-encoder functional regularization, we follow the details in Appendix D.1 to get the functional approximation of the model. For each model, we obtain a $\mathbb{R}^{10000 \times 10}$ matrix with softmax values for 10 target classes for each of the 10000 test points. We perform 100 independent runs for each method (with and without the functional regularization) obtaining 200 functional approximation vectors in $\mathbb{R}^{100,000}$. We visualise these vectors in 2D using the Isomap [50] algorithm [3] in Figure 3(b).

**Results** From Figure 3(a), we observe that the test accuracy of end to end training is inferior to that of using functional regularization with unlabeled data across a variety of labeled data sizes. We observe that the difference in the test accuracy between the two methods is highest when the amount of labeled data available is small and the performance gap decreases as the amount of labeled data increases, as predicted by our theory.

Figure 3(b) visualizes the functional approximation learned by the model. It shows that when using the denoising auto-encoder functional regularization, the learned functions stay in a smaller functional space, while they are scattered when using end to end training. This is in line with our empirical observations on controlled data, and our intuition for the theoretical analysis: pruning the representation hypothesis space via functional regularization translates to a compact functional space.

### E.2 Sentence Pair Classification

We consider the application of sentence pair classification using the Microsoft Research Paraphrase Corpus [15][4] which has sentence pairs with annotations of whether the two sentences are semantically equivalent. This dataset has approximately 3.7k and 1.7k sentence pairs in the train and test splits respectively. Here we specifically choose the MRPC dataset as it has a smaller size of labeled training data in comparison to most NLP datasets. To show the empirical benefits of using unlabeled data in addition to the limited train data available, we use a pre-trained BERT [13] language model. BERT, based on a transformer architecture, has been pre-trained using a masked token self-supervision task which involves masking a portion of the input sentence and using BERT to predict the masked tokens.

| Train Data Size | 200 | 500 | 1000 | 2000 | 3668 |
|---|---|---|---|---|---|
| BERT-FT | **68.1 / 80.6** | **71.0 / 80.6** | **72.7 / 81.8** | **74.9 / 82.4** | **80.3 / 85.7** |
| Random | 64.1 / 74.8 | 64.7 / 75.66 | 67.0 / 80.1 | 68.9 / 79.0 | 68.9 / 79.3 |
| Random-$\ell_1$ | 54.7 / 65.1 | 62.6 / 75.5 | 63.6 / 76.7 | 63.4 / 76.6 | 66.3 / 79.6 |
| Random-$\ell_2$ | 65.3 / 78.6 | 66.4 / 79.7 | 65.3 / 78.6 | 65.0 / 78.4 | 66.5 / 79.9 |

Table 1: Performance of fine-tuning pre-trained BERT (BERT-FT) and end-to-end training of a randomly initialised BERT on varying the **MRPC** training dataset size. Metrics are reported in the format Accuracy/F1 scores on the test dataset. The training data size is 3668 sentence pairs.

This pre-training is done over a large text corpus ($\sim$ 2 billion words) and hence we can view the pre-trained BERT, under our framework, as having already pruned a large fraction of the hypothesis space of $\mathcal{H}$ for learning the representation on the input text.

**Experimental Details**  We compare the performance of fine-tuning the pre-trained BERT with training a randomly initialised BERT from scratch. For the latter, we use three different loss formulations to further study the benefits of regularization on the text representation being learnt: (i) the Cross-Entropy loss $\mathcal{L}_{CE}$ on the predicted output (ii) $\mathcal{L}_{CE}$ along with a $\ell_1$ norm penalty on the representation (i.e, the 768-dimensional representation from BERT corresponding to the [CLS] token) (iii) $\mathcal{L}_{CE}$ along with a $\ell_2$ norm penalty on the representation. We refer to these three different loss formulations as Random, Random-$\ell_1$ and Random-$\ell_2$ respectively for notational simplicity. We want to study how varying the labeled data can impact the performance of different training methods. We present the results in Table 1. We use the 12-layer BERT Base uncased model for our experiments with an Adam optimizer having a learning rate $2e^{-5}$. We perform end to end training on the training data and tune the number of fine-tuning epochs. We report the accuracy and F1 scores as the metric on the test data averaged over 3 runs. When randomly initialising the weights of BERT, we use a standard normal distribution with mean 0 and standard deviation of 0.02 for the layer weights and set all the biases to zero vectors. We set the layer norms to have weights as a vector of ones with a zero vector as the bias. When adding the $l_p$ penalty on the BERT representations on randomly initialising the weights, we choose an appropriate weighting function $\lambda$ to make the training loss a sum of the cross entropy classification loss and $\lambda$ times the $l_p$ norm of the BERT representation. The $\lambda$ is chosen $\in [10^{-3}, 10^3]$ by validation over a set of 300 data points randomly sampled from the training split. We use the huggingface transformers repository [5] for our experiments.

**Results**  From the table, we observe that the performance of training BERT from pre-trained weights is better than the performance of training the BERT architecture from randomly initialised weights. When viewed under our framework, this empirically shows the benefits of using a learnable regularization function over fixed functions like the $\ell_1$ or $\ell_2$ norms of the representation.

On increasing the training data size, we observe that the performance of all the four training modes increases. However, we can see that the performance improvement of Random, Random-$\ell_1$ and Random-$\ell_2$ is marginal when compared to the improvement in BERT Fine-tuning. The latter can be attributed to the fact that the pre-trained weights of BERT are adjusted by specialising them towards the target data domain. To support this, in addition to Table 1, we also experimented by keeping the BERT weights fixed and only training the classifier. We observe that under such a setting, when we use a small training set, the model is unable to converge to a model different from the initialisation as similarly observed by [28]. This means that the learning indeed needs searching over a set of suitable hypotheses. Thus, we can conclude that unlabeled data helps in restricting the search space, and a small labeled data set can find a hypothesis suitable for the target domain data within the restricted search space, consistent with our analysis.

## Footnotes

[2]The optimal product of $V$ and $W$ should span the subspace of the top $r$ eigenvectors of $\Sigma$. But note that there are different pairs of $V$ and $W$ which can achieve the same product.

[3] The t-SNE algorithm focuses more on neighbour distances by allowing large variance in large distances, while Isomap approximates geodesic distance via shortest paths thereby working well in practice with larger distances. Compared to the controlled data experiments where the functional approximation lies in $\mathbb{R}^{1000}$, the functional approximation for Fashion-MNIST lies in $\mathbb{R}^{100,000}$, thereby visualizing better via Isomap than t-SNE.

[4] https://www.microsoft.com/en-us/download/details.aspx?id=52398

[5]https://github.com/huggingface/transformers