[Reviews · NeurIPS 2020]

Review 1

Summary and Contributions: The authors suggest a new theoretical framework to study unsupervised learning. The main idea, inspired by popular approaches to supervised learning, is to consider a collection of functions (representations), and searching a representation that minimizes some loss. This formulation is shown to capture various representation learning tasks, such as auto-encoders and self-supervised learning. The authors also prove some basic sample complexity results.

Strengths: The suggested framework is quite natural

Weaknesses: The extension suggested in the paper is quite straight forward, and similar suggestions have been made in the past (even Vapnik's framework in his book shown to capture several representation learning problems). The proofs are also based on standard techniques.

Correctness: yes

Clarity: yes

Relation to Prior Work: yes

Reproducibility: Yes

Additional Feedback:


Review 2

Summary and Contributions: ====== Post-rebuttal comments Thank you for the response. I am happy with the explanations and will increase my score, thus recommending the paper for acceptance. ====== The paper provides a theoretical background for learning tasks that combine two steps: i) representation learning (e.g., via auto-encoders or self-supervised learning) and ii) supervised learning with instances represented via the features learned in step i). The assumption is that in addition to labelled examples the algorithm has access to unlabelled instances. The first step learns a representation function h(x) that belongs to some hypothesis space H. The loss of a representation function is defined with respect to a set of hypothesis that are called regularizers. In particular, for a set of regularization functions G and a sampling distribution U over the instance space one defines the loss as (see Eq. 1) L(h, g, U) = E_{x ~ U} [ L(h, g, x) ]. For example, in auto-encoders L(h, g, x) = || x - g(h(x)) ||^2. The regularization loss is defined by minimizing with respect to G (see Eq. 2) L(h, U) = min_{g \in G} L(h, g, U). The setting of interest to this paper is best described in Eq. (4) min_{f \in F, h \in H} L(f, h, S) subject to L(h, U_e) <= t , where H is the hypothesis space for the representation function h and F is the hypothesis space for the supervised learning problem. Here S denotes an empirical sample of labelled examples and U_e a sample of unlabelled instances. The idea is that hypothesis in G act as regularizers in representation learning and that in this way it is possible to reduce the sample complexity of the supervised learning problem by exploiting information in unlabelled data. Section 4.2 provides sample complexity bounds for three different cases: i) unlabelled data and marginal distribution are overlapping, hypothesis spaces are finite, and it is a well-specified problem, ii) unlabelled data and marginal distribution overlap, infinite hypothesis classes, and misspecified model, iii) domain shift when it comes to unlabelled data, infinite hypothesis classes, and misspecified models. Section 5 casts concrete problems in this framework and provides some empirical analysis.

Strengths: I think this is an interesting problem and a nice idea that could provide some insights into potential gains as a result of exploiting information from unlabelled data.

Weaknesses: Perhaps to have more focus on cases when this will not work. For example, it is well known from previous analysis of semi-supervised learning that adding unlabelled data does not always help (to my recollection there were few papers with constructive proofs). It is not obvious where this fits into the assumptions and sample complexity bounds. Chapter 4 in Chapelle et al. (2006) might be helpful for the required assumptions in semi-supervised learning.

Correctness: In the appendix, the proof of Theorem 1, there is one thing that is not completely clear to me and it might be good to address it in the rebuttal (it might also be that I need to have a better look). I see Eq. (15) and understand the union bound argument. However, I am not really following the sentence after that in lines 594-595. In particular, the predicates are not completely clear to me. The way I understand Eq. (15) it should be P ( L_r(h, g, D_x) >= e_0 and L_r(h, g, U) = 0 ) <= ... I am having problems with conclusion: L_r(h, g, D_x) <= e_0 and L_r(h, g, U) = 0. Did you have some other predicate in mind? I will have a better look at the appendix during the discussions.

Clarity: The paper is well written and easy to follow. I have had no problems with the main part. The structure of the work is also good, with proper instantiations of the framework.

Relation to Prior Work: The work might benefit from a more thorough review of previous results in semi-supervised learning.

Reproducibility: Yes

Additional Feedback: My score is not final and will be determined by the rebuttal and discussion afterwards.


Review 3

Summary and Contributions: This paper evaluated the error of learning in a combination of supervised and unsupervised learning. Specifically, the error evaluation of supervised learning was performed in a situation where the class of learnable functions was constrained from unlabeled data. Theoretically, the number of samples required to achieve a given error level was analyzed. The analysis was by standard number of coverings, and bounding was given that could be used in infinite hypothesis sets.

Strengths: It is an ambitious attempt to analyze the theory of an important problem. A unified framework encompassing a variety of models provides generic results.

Weaknesses: One point for improvement is that the discussion is very abstract and does not provide much insight into the specific issues. First, it is very straightforward to say that the number of samples needed is determined by the maximum of covering numbers of the two learning hypotheses. If there was an interaction between the two learners, the discussion would go on, but at present, such an effect is not analyzed. So it does not provide a new intuition for the problem setting. Second, the discussion using the covering numbers is not very effective in practice. When investigating methods using neural networks such as VAE, it is known that evaluation of the covering numbers based on uniform convergence does not explain much performance [1]. Therefore, there is little validity for this method to explain actual representation learning. [1] Nagarajan+ Uniform convergence may be unable to explain generalization in deep learning, Neurips 2019.

Correctness: Good enough, as far as I've checked.

Clarity: Not bad, but it wasn't easy to read.

Relation to Prior Work: Good enough, as far as I've checked.

Reproducibility: Yes

Additional Feedback:


Review 4

Summary and Contributions: The paper provides an unified framework for analysis of several representation learning methods using unlabeled data. More precisely, the authors impose a regularization on the learned representation using a learnable function and derive sample complexity bounds for three different assumptions of data and hypothesis classes. The paper also shows that functional regularization can reduce the size of the labelled set needed by pruning the hypothesis class.

Strengths: The paper is well written and all the theoretical claims are well established using concrete theorems. The paper presents the claims for three different settings – same domain (finite hypothesis class), same domain (infinite hypothesis class) and the more practical different domain (infinite hypothesis class). Examples on auto-encoder and masked self-supervision also helps understand the claims in a more reachable manner. The authors also performed experiments on a sample dataset for both the auto-encoder and the masked self-supervision cases. From the findings, it can be clearly seen that functional regularization greatly helped reducing the error compared to a standard end-to-end learning paradigm.

Weaknesses: Even though it is understood that the paper is more theoretical in nature, the experimental section feels short. Experiment on a real-world dataset (like MNIST) would have helped understand the effect of the functional regularization claim concretely. In line 135, the l_c(f(h(x)), y) function is undefined. In line 179, “Recall that standard analysis shows … same error guarantee”. This analysis is not shown explicitly in the preceding pages of the paper.

Correctness: Probably so

Clarity: More or less clear

Relation to Prior Work: Need to do a thorough search. Some related work missed: 1. On mutual information maximization for representation learning - Michael Tschannen, Josip Djolonga, Paul K. Rubenstein et. al., ICLR 2020. 2. Few-Shot Learning via Learning the Representation, Provably – Simon S. Du, Wei Hu, Sham M. Kakade, Jason D. Lee, Qi Lei, Feb 2020, arXiv

Reproducibility: No

Additional Feedback: Authors should go through entire doc, to identify a few typos.

[Author Response · NeurIPS 2020]

We individually respond to the questions and concerns raised by reviewers about our work below:

**Reviewer 1:**
- **"The framework is natural but straight-forward"**: We are glad that R1 thinks that our framework is natural. However, inspite of being simple, such a unified view has not been applied to explain popular representation learning approaches, especially recent ones such as masked self-supervision and VAE. We believe that formulating these approaches theoretically and analysing their sample complexity bounds is an important problem in the domain of representation learning and can lead to insights. We view the simplicity of the problem formulation to be a strength rather than a weakness.

- Additionally, we present two concrete applications of the framework, with empirical support thereby showing the validity of our framework. This shows that our general framework is effective and doesn't lead to vacuous bounds.

**Reviewer 2:**
- **"Perhaps to have more focus on cases when this will not work"**: Our theorems and results do provide implications for cases when the auxiliary self-supervised task may not help the prediction task. For example consider Theorem 1, if $\mathcal{H}_{\mathcal{D}_X, L_r}(\epsilon_0)$ is not significantly smaller than $\mathcal{H}$, then using unlabeled data will not reduce the sample size of the labeled data much compared to only using the labeled data for prediction. Further, for the two concrete examples that we present in Section 5, the sample complexity bounds also indicate when the unlabeled data will not be very useful. Another possible reason that these representation learning approaches can fail is that the optimization does not provide a good solution. However, accounting for this is out of scope of this work. We will incorporate the suggestion and add more discussion about the related works pointed out.

- **Clarification about Equation 15**: This equation means that if we fix a pair $(h, g)$ with $L_r(h, g, D_x) \geq \epsilon_0$, then we have $P(L_r(h, g, U) = 0) \leq \delta/2|H||G|$. Since there are at most $|H||G|$ such $(h, g)$ pairs, by the union bound we have $P\big(\exists (h, g) \text{ s.t. } L_r(h, g, D_x) \geq \epsilon_0 \text{ and } L_r(h, g, U) = 0\big) \leq \delta/2$. Then with probability at least $(1 - \delta/2)$, there exists no $(h, g)$ such that $L_r(h, g, D_x) \geq \epsilon_0$ and $L_r(h, g, U) = 0$. This means that only those $(h, g)$ with $L_r(h, g, D_x) \leq \epsilon_0$ will have $L_r(h, g, U) = 0$.

**Reviewer 3:**
- **"Interaction between the two learners"**: Our sample complexity bounds do quantify the interaction between the two learners (from the hypothesis classes $\mathcal{H}$ and $\mathcal{G}$) by introducing the notion of $\mathcal{H}_{\mathcal{D}_X, L_r}$. This captures the effect that the representation learner over $\mathcal{G} \circ \mathcal{H}$ has on the prediction learner over $\mathcal{F} \circ \mathcal{H}$.

- **"Covering numbers are not very effective in practice"**: While we acknowledge the evidence that naively applying uniform convergence bounds may not result in good generalization/sample bounds for deep learning, these existing studies may not apply to the setting we consider in our paper. To the best of our knowledge, this evidence is specific to supervised learning without the auxiliary representation learning tasks, while our setting is with the auxiliary tasks. In particular, the mentioned paper "Uniform convergence may be unable to explain generalization in deep learning" does not apply directly to VAE or any other representation learning approach studied in our work.
Furthermore, our experimental results do correspond with our sample bounds in Section 5 which are based on uniform convergence. This is in contrast to the existing studies on supervised deep learning without auxiliary representation learning tasks. These results suggest considering a shifted view: "Uniform Convergence strikes back and can explain the generalization behavior of deep learning *with* auxiliary representation learning tasks". Why so? Without auxiliary representation learning tasks, it is generally believed that the optimization has an implicit regularization on the training, and hence uniform convergence fails to explain this. However with the auxiliary tasks, we conjecture that functional regularization restricts the learning dynamics to a smaller subset of hypotheses, on which the implicit regularization of the optimization is no longer significant, and thus the generalization can be explained by uniform convergence. This is an interesting open question and we leave it as future work.

**Reviewer 4:**
- **"Experiments on real world data would have helped"**: We have presented some experiments on real data in the appendix. Due to space restrictions, we only focus on experiments for the two concrete instantiations of our framework, which we believe can give fine-grained empirical evidence (e.g., how the sample bounds depend on $r$, etc.) for our analysis. We will move some experiments on real data to the main body in a future version of our paper.

- **Line 135**: The loss $l_c(f(h(x)), y)$ denotes the loss function for the prediction task, and has been defined on line 90.

- **Line 179**: By "standard analysis", we refer to the standard statistical learning theory argument for uniform convergence over a finite hypothesis class. We will make this explicitly clear in in a future version of our paper.

- **Related work**: Thanks for the suggestions! We will add them to the future version.

[Meta-Review · NeurIPS 2020]

This paper presents a unified framework for analyzing representational learning approaches that make use of unlabeled data for performing auxiliary tasks such as auto-encoders and masked self-supervision. The provided sample complexity bounds show that the auxiliary task provides a functional regularization that can prune the hypothesis space to reduce significantly the number of labeled examples sufficient for learning. The theory is confirmed experimentally on synthetic data. As I understand it, this work is the first to present a unified and natural framework to analyze the impact of unsupervised auxiliary tasks on generalization. Consequently, the novelty of the formulation and its applicability to algorithmic approaches of broad interest to practitioners outweighed the fact that some reviewers saw the technical contributions as rather straightforward. Finally, as suggested by a reviewer, we think that the authors should provide more discussion on the cases where the use of auxiliary tasks will not help (as it was done in the rebuttal).